# Factors enforcing the species boundary between the human pathogens *Cryptococcus neoformans* and *Cryptococcus deneoformans*

**Shelby J. Priest**[1], **Marco A. Coelho**[1], **Verónica Mixão**[2,3], **Shelly Applen Clancey**[1], **Yitong Xu**[4], **Sheng Sun**[1], **Toni Gabaldón**[2,3,5], **Joseph Heitman**[1]*

**1** Department of Molecular Genetics and Microbiology, Duke University Medical Center, Durham, North Carolina, United States of America, **2** Life Sciences Department, Barcelona Supercomputing Center, Barcelona, Spain, **3** Institute for Research in Biomedicine, Barcelona Institute of Science and Technology, Barcelona, Spain, **4** Program in Cell and Molecular Biology, Duke University Medical Center, Durham, North Carolina, United States of America, **5** Catalan Institution for Research and Advanced Studies (ICREA), Barcelona, Spain

* heitm001@duke.edu

**Data Availability Statement:** Raw sequence data has been deposited into the National Center for Biotechnology Information Sequence Read Archive under BioProject accession no. PRJNA626676.

## Abstract

Hybridization has resulted in the origin and variation in extant species, and hybrids continue to arise despite pre- and post-zygotic barriers that limit their formation and evolutionary success. One important system that maintains species boundaries in prokaryotes and eukaryotes is the mismatch repair pathway, which blocks recombination between divergent DNA sequences. Previous studies illuminated the role of the mismatch repair component Msh2 in blocking genetic recombination between divergent DNA during meiosis. Loss of Msh2 results in increased interspecific genetic recombination in bacterial and yeast models, and increased viability of progeny derived from yeast hybrid crosses. Hybrid isolates of two pathogenic fungal *Cryptococcus* species, *Cryptococcus neoformans* and *Cryptococcus deneoformans*, are isolated regularly from both clinical and environmental sources. In the present study, we sought to determine if loss of Msh2 would relax the species boundary between *C. neoformans* and *C. deneoformans*. We found that crosses between these two species in which both parents lack Msh2 produced hybrid progeny with increased viability and high levels of aneuploidy. Whole-genome sequencing revealed few instances of recombination among hybrid progeny and did not identify increased levels of recombination in progeny derived from parents lacking Msh2. Several hybrid progeny produced structures associated with sexual reproduction when incubated alone on nutrient-rich medium in light, a novel phenotype in *Cryptococcus*. These findings represent a unique, unexpected case where rendering the mismatch repair system defective did not result in increased meiotic recombination across a species boundary. This suggests that alternative pathways or other mismatch repair components limit meiotic recombination between homeologous DNA and enforce species boundaries in the basidiomycete *Cryptococcus* species.

**Funding:** This work was funded by NIH/NIAID F31 Fellowship 1F31AI143136-02A1 awarded to S.J.P. (https://www.niaid.nih.gov/) and NIH/NIAID R37 MERIT award AI39115-23, R01 grant AI50113-16, and R01 grant AI33654-04 awarded to J.H. (https://www.niaid.nih.gov/). J.H. is co-Director and Fellow of the CIFAR program Fungal Kingdom: Threats & Opportunities (https://www.cifar.ca/research/program/fungal-kingdom). Y.X. is supported by NIH training grant T32 GM007184 awarded to the Duke University Program in Cell and Molecular Biology (https://www.nigms.nih.gov/). V.M. and T.G. were supported by the European Union's Horizon 2020 Research and Innovation Program under the Marie Sklodowska-Curie grant agreement No. H2020-MSCA-ITN-2014-642095 (https://ec.europa.eu/programmes/horizon2020/en/h2020-section/marie-sklodowska-curie-actions). T.G. also acknowledges support from the CERCA Program/Generalitat de Catalunya (https://cerca.cat/en/), the Catalan Research Agency (AGAUR) SGR423 (http://agaur.gencat.cat/en/lagaur/), the INB Grant (PT17/0009/0023 – ISCIII-SGEFI/ERDF) (https://www.isciii.es/Paginas/Inicio.aspx), the European Union's Horizon 2020 Research and Innovation grant agreement ERC-2016-724173 (https://ec.europa.eu/programmes/horizon2020/en), and the Marie Sklodowska-Curie grant agreement No. H2020-MSCA-IF-2017-793699 (https://ec.europa.eu/programmes/horizon2020/en/h2020-section/marie-sklodowska-curie-actions). We also thank the Madhani Laboratory and NIH grant R01 AI100272 for the KN99α msh2Δ and fur1Δ deletion strains (https://www.niaid.nih.gov/). The funders had no role in study design, data collection and analysis, decision to publish, or preparation of the manuscript.

**Competing interests:** The authors have declared that no competing interests exist.

## Author summary

Several mechanisms enforce species boundaries by either preventing the formation of hybrid zygotes, known as pre-zygotic barriers, or preventing the viability and fecundity of hybrids, known as post-zygotic barriers. Despite these barriers, interspecific hybrids form at an appreciable frequency, such as hybrid isolates of the human fungal pathogenic species, *Cryptococcus neoformans* and *Cryptococcus deneoformans*, which are regularly isolated from both clinical and environmental sources. *C. neoformans* x *C. deneoformans* hybrids are typically highly aneuploid, sterile, and display phenotypes intermediate to those of either parent, although self-fertile isolates and transgressive phenotypes have been observed. One important mechanism known to enforce species boundaries or lead to incipient speciation is the DNA mismatch repair system, which blocks recombination between divergent DNA sequences during meiosis. The aim of this study was to determine if genetically deleting the DNA mismatch repair component Msh2 would relax the species boundary between *C. neoformans* and *C. deneoformans*. Progeny derived from *C. neoformans* x *C. deneoformans* crosses in which both parental strains lacked Msh2 had higher viability, and unlike previous studies in *Saccharomyces*, these *Cryptococcus* hybrid progeny had higher levels of aneuploidy and no observable increase in meiotic recombination at the whole-genome level.

## Introduction

The mixing of species through sexual reproduction can result in hybrid offspring. While hybridization can have beneficial consequences in some cases (e.g. hybrid vigor and the emergence of novel hybrid species), sexual reproduction between diverging lineages or different species is typically deleterious and results in hybrid progeny with reduced fitness or sterility [1–3]. Thus, mechanisms preventing such events, such as pre- and post-zygotic reproductive barriers, tend to be favored by natural selection. Pre-zygotic species barriers block the formation of a hybrid zygote, and in the event that a hybrid zygote forms, several post-zygotic barriers exist that inhibit the viability or fecundity of the hybrid [4]. Post-zygotic barriers include 1) gross chromosomal rearrangements that prevent effective meiotic recombination, 2) Bateson-Dobzhansky-Muller incompatibilities in which interactions between nuclear elements or between mitochondrial and nuclear factors are detrimental or lethal, and finally, 3) the mismatch repair (MMR) pathway, which has an important role in blocking meiotic recombination between diverged DNA sequences.

The MMR pathway was first identified in prokaryotes as a mechanism to repair replication errors or damage-induced mismatches in DNA [5]. The MMR pathway is highly conserved, playing similar roles in unicellular eukaryotes, such as *Saccharomyces* species, and in multicellular eukaryotes, including humans [6]. Prokaryotic and eukaryotic cells lacking functional MMR components typically have increased mutation rates and therefore display a hypermutator phenotype [7]. The MMR pathway also plays an additional role in maintaining species boundaries by inhibiting homeologous chromosome pairing and subsequent recombination during meiosis. The inability of chromosomes to properly and stably pair, and subsequently undergo recombination, can lead to chromosome nondisjunction during meiosis, resulting in high frequencies of aneuploidy. This aneuploidy can be lethal if a progeny fails to inherit one or more essential chromosomes or if a lethal combination of alleles is inherited. Rayssiguier et al. demonstrated that mutation of the MutL, MutS, or MutH MMR components relaxed the species boundary between *Escherichia coli* and *Salmonella typhimurium*, two bacterial species

whose genomes are 20% divergent, such that recombination during conjugational and transductional crosses increased up to 1,000-fold [8]. The involvement of MMR in recombination is conserved and has also been shown to enforce the species boundary between *Saccharomyces cerevisiae* and the closely related species *Saccharomyces paradoxus*, whose genomes are ~15% divergent [9]. Mutants lacking the eukaryotic MutS homolog, Msh2, or the MutL homolog, Pms1, produce hybrid *S. cerevisiae* x *S. paradoxus* progeny with higher rates of viability as well as approximately ten-fold increased frequencies of meiotic recombination [10,11].

Despite the numerous pre- and post-zygotic species barriers and the robust fitness defects associated with hybrids, hybridization occurs at an appreciable frequency. In an exciting recent study, researchers witnessed the origin and monitored the evolution of a novel finch species that arose through a hybridization event in the Galapagos Islands [12]. Hybridization has impacted recent human evolution as well, with several modern human populations having introgressed genomic regions from Neanderthals or Denisovans [13]. Ligers, the hybrid progeny of a male lion and female tiger, and mules, the hybrid progeny of a male donkey and female horse, are examples of hybrids that normally only occur through human intervention. Ligers are bred for their size, as they are larger than either parent (an instance of hybrid vigor), while mules are bred for their endurance, docile demeanor, and intelligence [14]. It is important to note however, that interspecific hybrids are often sterile [15].

Hybridization is also important in many microbial pathogens and model organisms. For instance, the diploid model organism *S. cerevisiae* is thought to have arisen following a whole-genome duplication that was a direct consequence of interspecies hybridization [16–18]. An instance of relatively recent hybridization is also thought to have led to the emergence of the novel widespread fungal plant pathogen species *Zymoseptoria pseudotritici*, originating from fusion between two diverged haploids followed by mitosis and meiosis to generate a recombinant haploid $F_1$ hybrid [19]. In recent years, the hybrid nature of several emerging human opportunistic pathogens has been uncovered [20,21], suggesting hybridization might be a mechanism underlying the emergence of novel pathogens [22].

Several *Cryptococcus* species are microbial human fungal pathogens and are responsible for over 200,000 infections in both immunocompromised and immunocompetent individuals annually [23]. Cryptococcal infections are associated with high mortality rates and occur globally. There are currently eight recognized species in the pathogenic *Cryptococcus* species complex that form two well-supported subgroups, the *Cryptococcus neoformans* species complex and the *Cryptococcus gattii* species complex, which consist of two and six species, respectively [24,25]. The present study focuses on the two members of the *C. neoformans* species complex: *C. neoformans* and *C. deneoformans*.

Previously, *C. neoformans* and *C. deneoformans* were recognized as a single species with two varieties and two serotypes: *C. neoformans* var. *grubii* (serotype A) and *C. neoformans* var. *neoformans* (serotype D) [24]. However, there is clear genetic evidence separating these two groups, and molecular phylogenetics along with whole-genome sequencing suggest they diverged ~18 million years ago [26–29]. There are also several phenotypes that differentiate *C. neoformans* and *C. deneoformans* as distinct species, such as differences in thermotolerance, capsular agglutination reactions, morphology during murine infection, and human disease manifestations and outcomes [30–34]. Of the pathogenic *Cryptococcus* species, *C. neoformans* and *C. deneoformans* are the two most commonly isolated species from clinical and environmental settings and both species serve as model pathogenic eukaryotic organisms [35]. Both species have bipolar mating-type systems in which a single mating-type (*MAT*) locus encodes either the *MAT***a** or *MAT*α mating-type allele. *C. neoformans* has only been observed to undergo bisexual reproduction, between cells of opposite mating types, while *C. deneoformans* is capable of both bisexual and unisexual mating, which occurs either between two cells of the

same mating type or via endoreplication [36]. Due to the large prevalence of *MAT*α strains isolated from clinical and environmental settings, *C. neoformans* and *C. deneoformans* are thought to largely reproduce through unisexual reproduction or asexually as haploids, with infrequent instances of bisexual reproduction in nature.

Despite differences between *C. neoformans* and *C. deneoformans*, these two groups produce hybrids in the laboratory and in nature. *C. neoformans* x *C. deneoformans* hybrids, also known as AD hybrids, make up ~7.5% of environmental isolates and up to 30% of clinical isolates in Europe and North America [37–40]. Spores produced by genetic crosses between *C. neoformans* and *C. deneoformans* isolates are known to have poor germination frequencies (~5%) relative to intraspecific crosses (~80% germination), and are typically highly aneuploid or heterozygous diploids [41]. This poor viability is likely due to a combination of gross chromosomal rearrangements between the two parental genomes along with ~15% sequence divergence between the parental species [27,29,42], leading to a compromised meiosis that produces genetically imbalanced meiotic progeny. *C. neoformans* x *C. deneoformans* hybrids also have unstable karyotypes, can be self-fertile, and display phenotypes intermediate of either parent, although instances of transgressive phenotypes (i.e. phenotypes that fall outside of the range between either parental phenotype) and hybrid vigor have been observed [41,43–45]. Several studies of *C. neoformans* x *C. deneoformans* hybrid genomes have utilized restriction fragment length polymorphism analysis or PCR with sequence-specific primers to assess hybrid genomes. Through these methods it has been demonstrated that while *C. neoformans* x *C. deneoformans* hybrids are heterozygous at most loci, some chromosomes seem to be recombinant, indicative of potential mitotic or meiotic recombination [44,46–48].

In the present study, we generated *C. neoformans* and *C. deneoformans* strains lacking Msh2 to determine if loss of MMR relaxed the boundary between these two species. As expected, *C. neoformans* and *C. deneoformans* *msh2*Δ mutants displayed hypermutator phenotypes. Hybrid progeny derived from genetic crosses in which both parental strains lacked Msh2 displayed increased germination frequencies compared to wild-type crosses and also exhibited phenotypes and genotypes in accordance with previous findings [41]. Several instances of genetic recombination were observed in hybrid progeny derived from both wild-type *C. neoformans* x *C. deneoformans* crosses and *msh2*Δ mutant crosses, although interestingly, increased frequencies of meiotic recombination were not observed in hybrid progeny derived from crosses involving *msh2*Δ mutants. Additionally, lower rates of loss of heterozygosity (LOH), higher rates of aneuploidy, and more instances of chromosome breaks and *de novo* telomere addition were observed in hybrid progeny from *msh2*Δ mutant crosses. These results suggest that although Msh2 plays a role in the viability of hybrid progeny, other pathways and mechanisms are responsible for blocking homeologous meiotic recombination in *Cryptococcus*.

## Results

### *C. deneoformans* *msh2*Δ mutants are hypermutators

To assess the role of the MMR pathway in maintaining species boundaries in *Cryptococcus*, we first determined if Msh2 plays a similar role in DNA MMR in *C. deneoformans* as in other fungi. Deletion mutants lacking *MSH2* were generated via biolistic transformation and homologous recombination in the *C. deneoformans* JEC20**a** genetic background (S1A Fig). After transformation, mutants were selected on medium containing nourseothricin, and PCR was employed to confirm that the deletion allele had replaced the wild-type *MSH2* allele at its endogenous locus (S1B Fig).

Following isolation and confirmation of the desired *msh2*Δ mutants, we determined if the mutants displayed hypermutator phenotypes similar to those observed in *msh2* mutants of

other fungi [49,50]. To assess mutation rate, fluctuation assays were performed on either YPD medium supplemented with a combination of rapamycin and FK506 at 37°C (where calcineurin, the target of FKBP12-FK506, is essential) or YNB medium supplemented with 5-fluoroorotic acid (5-FOA) at 30°C. Resistance to the combination of rapamycin and FK506 is mediated by mutations in their common target, FKBP12, which is encoded by the gene *FRR1*, while resistance to 5-FOA arises following loss-of-function mutations in *URA5* or *URA3*, genes encoding enzymes involved in the *de novo* pyrimidine biosynthesis pathway. In this analysis, the progenitor strain of the genetic deletion mutants, JEC20**a**, served as the negative control, and a *C. neoformans msh2Δ* mutant in the KN99α genetic background from the *Cryptococcus* deletion mutant collection [51] served as the positive control. Although fluctuation assays do not provide genome-wide mutation rates, the assays allow us to compare the mutation rates at two unique coding loci of the JEC20**a** *msh2Δ* mutants to those of the controls. On the rapamycin and FK506 antifungal drug combination, the *msh2Δ-1*, *msh2Δ-2*, *msh2Δ-3*, and *msh2Δ-4* mutants exhibited hypermutator phenotypes with significantly higher mutation rates (*msh2Δ-1*: $1.02 \times 10^{-6}$ (95% confidence interval (CI) $7.76 \times 10^{-7}$–$1.28 \times 10^{-6}$); *msh2Δ-2*: $1.30 \times 10^{-6}$ (95% CI $1.04 \times 10^{-6}$–$1.59 \times 10^{-6}$); *msh2Δ-3*: $1.16 \times 10^{-6}$ (95% CI $9.01 \times 10^{-7}$–$1.45 \times 10^{-6}$); and *msh2Δ-4*: $1.16 \times 10^{-6}$ (95% CI $9.01 \times 10^{-7}$–$1.45 \times 10^{-6}$) mutations per cell per generation) than the parental JEC20**a** strain ($8.59 \times 10^{-8}$ (95% CI $4.84 \times 10^{-8}$–$1.31 \times 10^{-7}$) mutations per cell per generation) (Fig 1A). Mutation rates of three of the four independent *msh2Δ* mutants (*msh2Δ-2*, *msh2Δ-3*, and *msh2Δ-4*) were also significantly higher than the mutation rate of the parental strain, JEC20**a**, on 5-FOA (S2A Fig). Interestingly, the *msh2Δ-1* mutant failed to produce any 5-FOA resistant isolates during the fluctuation assay, which was explained by a single base deletion in a homopolymeric nucleotide run, causing a frameshift in *FUR1*, which encodes a uracil phosphoribosyl transferase involved in the pyrimidine salvage pathway. This mutation led to cross resistance to the antifungal drug 5-fluorouridine (5FU) and the clinically relevant antifungal drug 5-fluorocytosine (5FC) (S2B-S2D Fig) (see Materials and Methods for further details) [52].

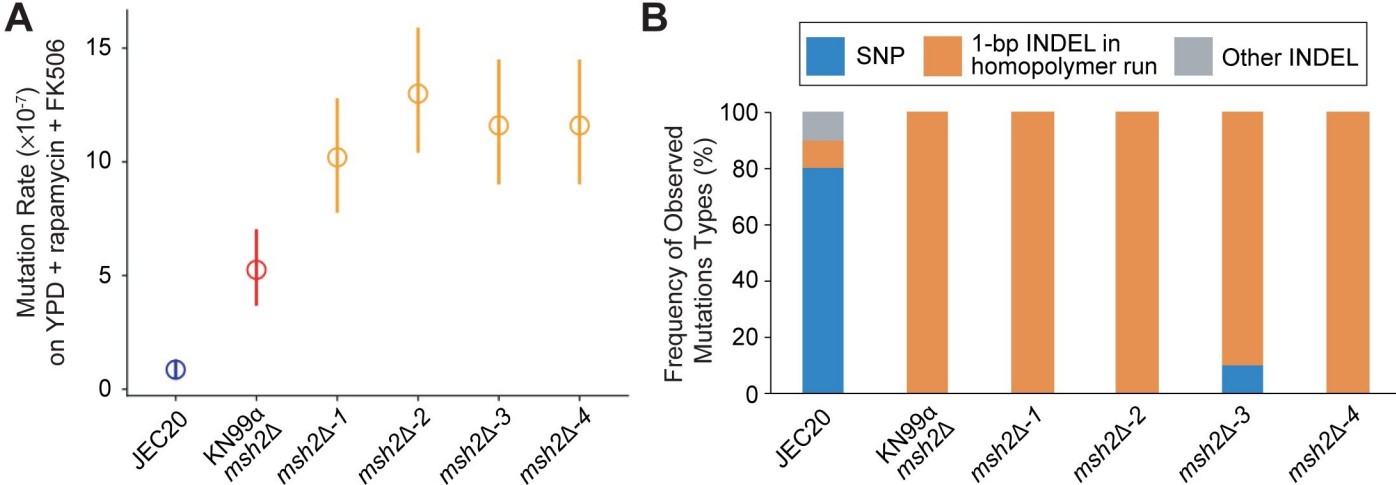

**Fig 1. Hypermutator phenotypes of JEC20a *msh2Δ* mutants.** (A) Fluctuation analysis on YPD + rapamycin + FK506 medium was performed to quantify the mutation rates (number of mutations per cell per generation) of four independent JEC20**a** *msh2Δ* mutants: *msh2Δ-1*, *msh2Δ-2*, *msh2Δ-3*, and *msh2Δ-4*. The JEC20**a** progenitor strain in which the *msh2Δ* mutants were constructed served as the negative control and an *msh2Δ* mutant in the KN99α genetic background served as the positive control. Points indicate mean mutation rates and error bars indicate 95% confidence intervals for the mean; 10 independent replicates of each strain were included in mutation rate calculation. **(B)** Spectra of mutations identified through sequencing of the *FRR1* gene in rapamycin + FK506-resistant colonies from fluctuation analysis conducted in panel A. For JEC20**a** n = 9, for all other strains, n = 10.

Insertion/deletion mutations (INDELs) in homopolymeric nucleotide runs are a hallmark mutation pattern of *msh2Δ* mutants [49,53]. To determine if the four independent JEC20**a** *msh2Δ* mutants also displayed similar mutation patterns, *FRR1*, the gene encoding FKBP12 (the common target of FK506 and rapamycin), was PCR amplified and mutations were identified through Sanger sequencing. This analysis revealed single base pair INDELs in homopolymeric nucleotide runs within *FRR1* in all ten independent colonies analyzed for the *msh2Δ-1*, *msh2Δ-2*, and *msh2Δ-4* strains and nine of ten independent colonies analyzed for the *msh2Δ-3* strain (Fig 1B). This is significantly different from the frequency of 1-bp INDEL mutations observed in the wild-type JEC20**a** parental strain, in which only one of nine colonies had a 1-bp INDEL mutation in a homopolymer run ($p<0.001$, Fisher's exact test). As anticipated, the KN99α *msh2Δ* mutant produced resistant colonies with 1-bp INDEL mutations in homopolymer runs in *FRR1* in all (10/10) colonies analyzed (Fig 1B).

## Progeny from *msh2Δ* hybrid genetic crosses display increased viability

Interspecific crosses involving MMR-deficient *Saccharomyces* strains produce progeny with increased viability [10,11]. To determine if a similar increase in viability would be observed in *Cryptococcus*, genetic crosses were conducted between *C. neoformans* and *C. deneoformans* wild-type strains as well as corresponding *msh2Δ* mutants. A cross between H99α (a laboratory standard *C. neoformans* reference strain) and JEC20**a** served as a control for the *msh2Δ* hybrid crosses. H99α x JEC20**a** interspecific crosses produced robust mating structures, including hyphae, basidia, and basidiospores (Fig 2A). Interestingly, bilateral crosses in which both parental strains lacked *MSH2* produced similarly abundant hyphae as those produced by the H99α x JEC20**a** cross, but produced a significantly greater number of bald basidia and therefore fewer basidiospore chains (34%, 33%, and 63% bald basidia on average for the wild-type, unilateral, and bilateral *msh2Δ* hybrid crosses, respectively, $p<0.001$, one-way ANOVA, Tukey's HSD) (Fig 2A and 2B and S1 Table).

Basidiospores from wild-type, unilateral *msh2Δ* x wild type, and bilateral *msh2Δ* x *msh2Δ* *C. neoformans* x *C. deneoformans* genetic crosses involving each of the four independent JEC20**a** *msh2Δ* mutants and the KN99α *msh2Δ* strain were randomly dissected via micromanipulation onto nutrient-rich YPD medium. The spores were allowed to germinate for up to three weeks on YPD medium at room temperature, and total germination frequencies were calculated (Fig 2C and S2 Table). Dissected basidiospores from wild-type H99α x JEC20**a** crosses germinated at a frequency of 5.8%, which is similar to the previously published frequency of 5% [41]. Basidiospores from unilateral crosses in which only one parent lacked *MSH2* germinated at a frequency of 7.5% (Fig 2C and S2 Table). Hybrid progeny from bilateral crosses involving *msh2Δ* mutants in both parents displayed significantly increased germination frequencies compared to the progeny from unilateral and wild-type crosses, with an average germination frequency of 29% ($p<0.001$, one-way ANOVA, Tukey's HSD). Progeny germination frequencies were as high as 41% for some bilateral *msh2Δ* x *msh2Δ* sexual crosses (S2 Table), nearing the expected upper germination frequency limit for *C. neoformans* x *C. deneoformans* crosses based on the presence of one large reciprocal chromosomal translocation between *C. neoformans* H99α and *C. deneoformans* JEC21α, a strain congenic to JEC20**a** with the exception of the *MAT* locus [29,42].

The germination frequencies of progeny from the hybrid *C. neoformans* x *C. deneoformans* crosses were much lower than germination frequencies from intraspecific crosses (S3 Fig and S2 Table). For instance, progeny from an intraspecific *C. neoformans* cross between the wild-type strains H99α and KN99**a** germinated at an average frequency of 83%. In a unilateral *msh2Δ* *C. neoformans* intraspecific cross, progeny germinated at an average frequency of 86%, while progeny from a bilateral *msh2Δ* *C. neoformans* intraspecific cross germinated on average

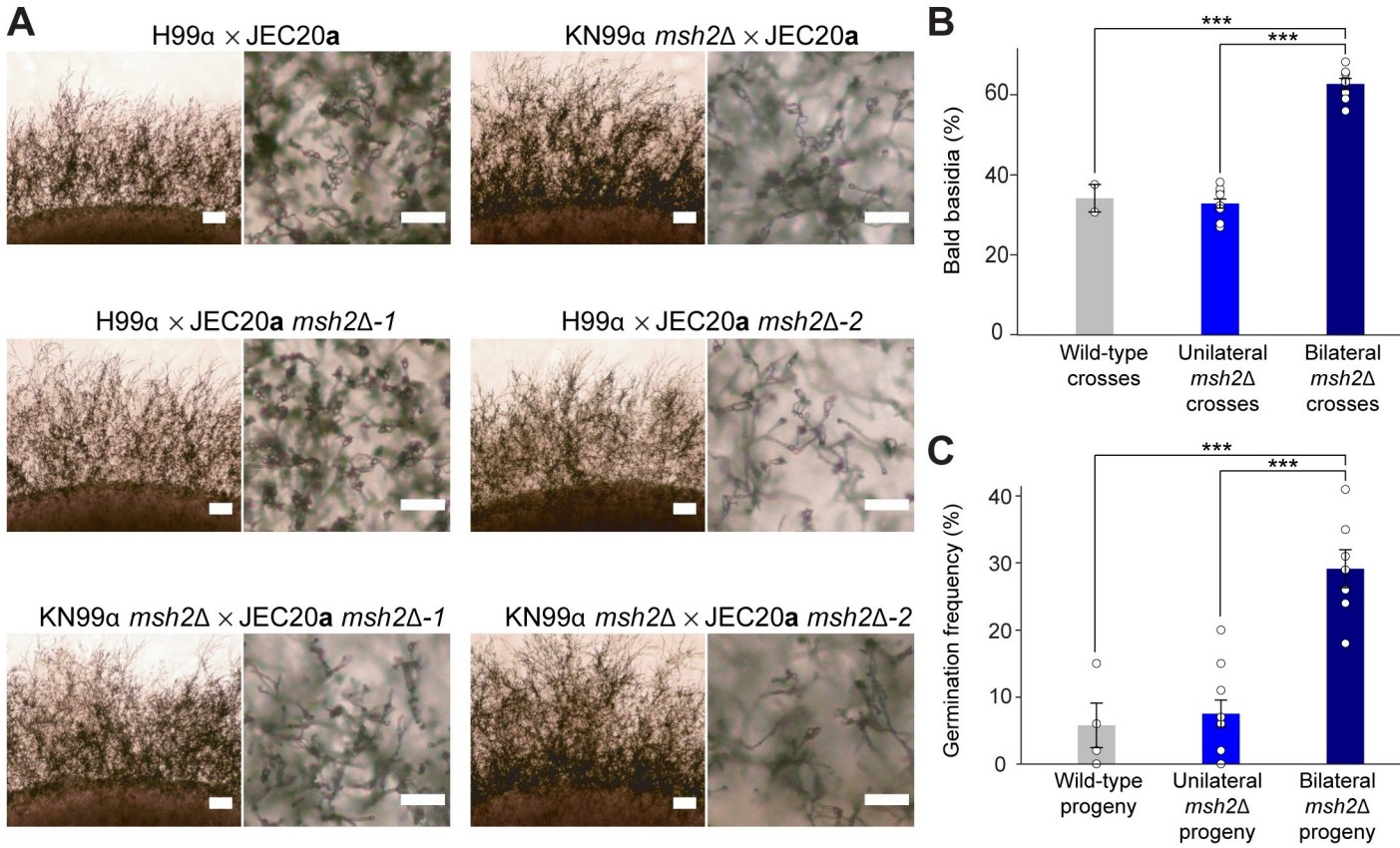

**Fig 2. Mating structure formation in *C. neoformans* x *C. deneoformans* genetic crosses and germination frequencies of hybrid progeny. (A)** *C. neoformans* x *C. deneoformans* genetic crosses produced robust hyphal filamentation after 6 days of incubation in dark conditions on MS medium. Scale bars in colony border images (left image of each set) represent 200 μm. Scale bars in basidia and basidiospore morphology images (right image of each set) represent 50 μm. **(B)** Average frequencies of basidia lacking basidiospores, or bald basidia, in *C. neoformans* x *C. deneoformans* genetic crosses. Error bars represent standard error of the mean. Statistical significance was determined with a one-way ANOVA and Tukey's post hoc test. **(C)** Mean germination frequencies of *C. neoformans* x *C. deneoformans* hybrid progeny in wild-type, unilateral, and bilateral *msh2Δ* genetic crosses. Error bars represent standard error of the mean and statistical significance was determined by one-way ANOVA followed by Tukey's post hoc test. *** indicates $p < 0.001$.

only 71% of the time, which was significantly lower than either the unilateral *msh2Δ* or wild-type *C. neoformans* intraspecific progeny ($p < 0.05$, one-way ANOVA, Tukey's HSD). Similarly, progeny from wild-type (JEC20**a** x JEC21α) and unilateral *msh2Δ* *C. deneoformans* intraspecific crosses had high average germination frequencies (78% average). In contrast, progeny from bilateral *msh2Δ* *C. deneoformans* intraspecific crosses germinated only 36% of the time, a significantly lower frequency compared to the wild-type and unilateral crosses ($p < 0.001$, one-way ANOVA, Tukey's HSD) (S3B Fig and S2 Table). The decreases in germination frequencies in progeny from bilateral *msh2Δ* intraspecific crosses is likely due to the high mutation rates associated with loss of Msh2, as has been observed in other studies, although this effect appears to be more pronounced in *C. deneoformans* than in *C. neoformans* [10,54].

## Hybrid progeny are capable of hyphal growth and sporulation on nutrient-rich medium and display phenotypes and genotypes typically associated with *C. neoformans* x *C. deneoformans* hybrids

During the prolonged incubation period in which dissected progeny were allowed to germinate, YPD agar germination plates were kept sealed with parafilm on the benchtop.

Surprisingly, after 14 days of incubation, aerial filaments began to emerge from the periphery of several germinated hybrid progeny. Through microscopic analysis we found that these progeny had produced structures resembling those observed during sexual reproduction under mating-inducing conditions. Continued incubation and additional microscopic analysis revealed that after approximately three weeks of incubation, a number of these progeny produced basidia and sparse basidiospores under conditions previously not known to support sexual reproduction for any *Cryptococcus* species or strains (rich medium, incubated in the light in sealed plates) (Fig 3). This phenotype was measured across all hybrid progeny, parental strains, and standard laboratory reference strains. Although there was no significant difference between the ability to produce hyphae on YPD between the progeny from the different types of hybrid crosses (one-way ANOVA), progeny from bilateral *msh2Δ* hybrid crosses tended to be able to produce hyphae more often on average than progeny from wild-type or unilateral *msh2Δ* hybrid crosses (45% compared to 25% or 19%, respectively) (Fig 3B and S3 Table). Interestingly, no parental strains, progeny from intraspecific crosses, or common laboratory reference strains were able to produce hyphae on YPD except for XL280α, a hyper-filamentous *C. deneoformans* strain [55].

The hybrid progeny from both wild-type *C. neoformans* x *C. deneoformans* crosses, as well as those from unilateral and bilateral *msh2Δ* crosses, exhibited typical *Cryptococcus* hybrid genotypes: high levels of aneuploidy and inheritance of the *MATa* and *MATα* mating-type alleles from both parents [41,45]. With sequence-specific primers for the gene *STE20*, which exists as two mating-type specific alleles and encodes a kinase involved in the pheromone response signaling cascade, the mating types of the hybrid progeny were determined. Nearly all progeny inherited and maintained both the *STE20α* allele from the *C. neoformans* parent and the *STE20a* allele from the *C. deneoformans* parent, which was expected due to the diploid genome characteristic of *C. neoformans* x *C. deneoformans* hybrids (S4 Fig) [41,45]. Based on flow cytometry, the ploidy of the majority of the hybrid progeny was diploid or aneuploid (S5 Fig). Three of the 27 hybrid progeny for which whole-genome sequencing (WGS) was obtained (progeny YX1, YX3, and YX6) were, however, estimated to be close to haploid, and all three were from a wild-type *C. neoformans* x *C. deneoformans* cross (S5 Fig). These results are in stark contrast to the FACS analysis results for 38 intraspecific progeny from the *C. neoformans* and *C. deneoformans* wild-type, unilateral *msh2Δ*, and bilateral *msh2Δ* crosses, all of which were estimated to be haploid, with one exception: KN99α *msh2Δ* x KN99a progeny 5 appeared diploid (S6 Fig).

Although the majority of naturally occurring *C. neoformans* x *C. deneoformans* hybrid strains are not self-fertile, *C. neoformans* x *C. deneoformans* hybrids produced under laboratory conditions, similar to those presented in this study, are often self-fertile [41,43]. This fertility is characterized by the ability to produce hyphae when incubated alone on mating-inducing media, such as MS agar plates, at room temperature in the dark. All but one of the hybrid progeny assessed produced hyphae on MS medium, and many progeny were also capable of producing basidia and basidiospores (S7 Fig). The only progeny that was not self-fertile (YX3) had lost the *C. deneoformans* *MATa* locus (S4 and S7 Figs). Interestingly, progeny YX1, which only inherited the *C. deneoformans* *MATa* locus but not the *C. neoformans* *MATα* locus, was self-fertile (S4 and S7 Figs). While the *C. deneoformans* JEC20a parent is not self-fertile, other *C. deneoformans* *MATa* strains are self-fertile [56], and genetic mechanisms underlying this fertility may be similar to those observed in YX1.

## Hybrid progeny from *msh2Δ* genetic crosses are highly aneuploid or diploid

Following isolation and characterization of hybrid progeny from the wild-type and unilateral and bilateral *msh2Δ* mutant crosses, we generated WGS data for 27 hybrid progeny. The

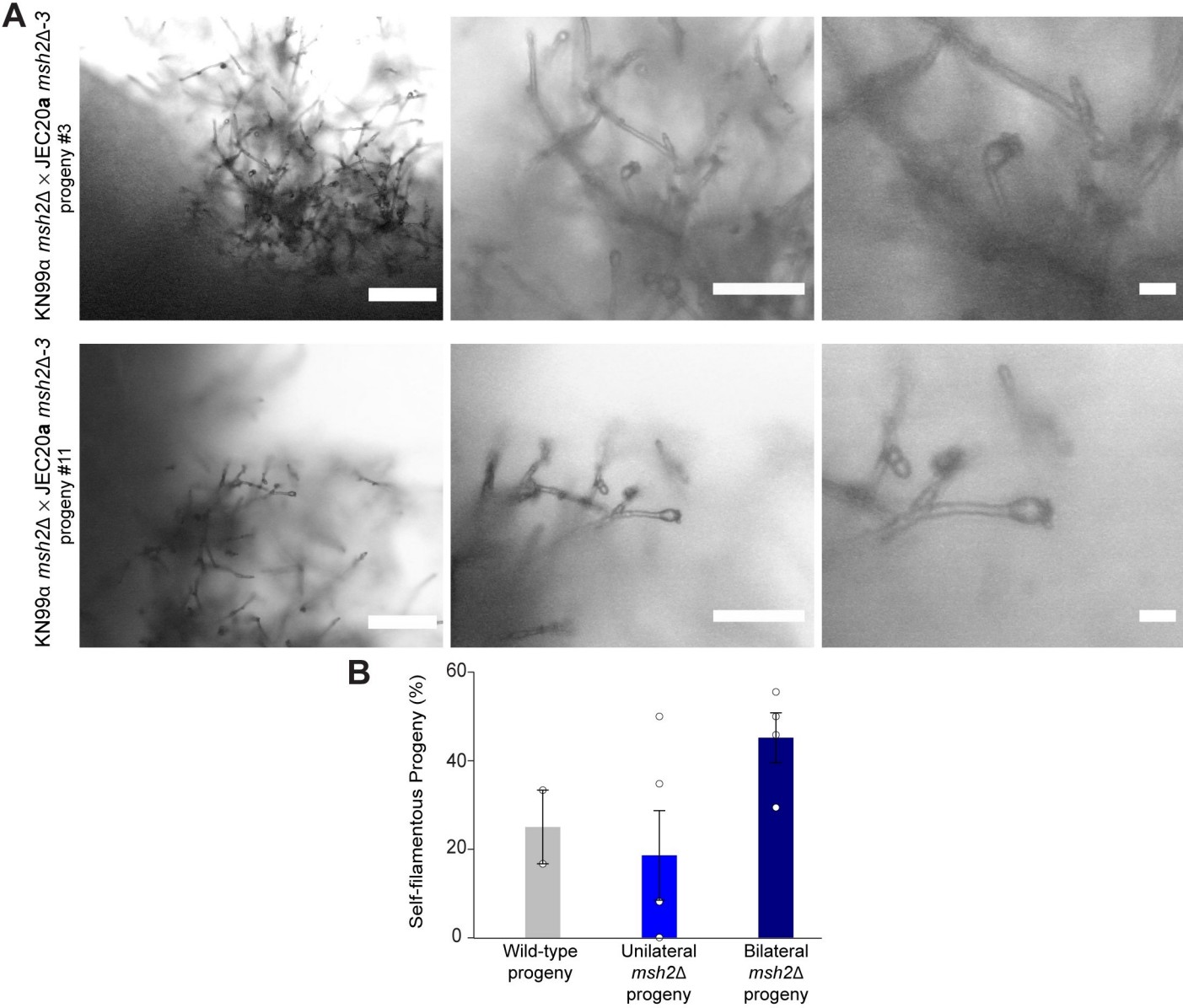

**Fig 3. Unique phenotypes of *C. neoformans* x *C. deneoformans* hybrid progeny.** (A) Sexual reproduction structures, including hyphae, basidia, and basidiospores, produced by *C. neoformans* x *C. deneoformans* hybrid progeny derived from bilateral *msh2Δ* genetic crosses following dissection and micromanipulation onto YPD agar medium. Sexual structures were formed on YPD agar plates sealed with parafilm and incubated in light conditions at room temperature. From left to right, scale bars on microscopy images represent 100 μm, 50 μm, and 10 μm. (B) Frequency of hybrid progeny capable of producing hyphae on YPD agar medium, at room temperature, in light, sealed conditions. Error bars represent standard error of the mean. One-way ANOVA identified no statistically significant differences between the three groups.

chromosomal composition of the hybrid progeny, along with the identification of potential sites of meiotic recombination, LOH events, and heterozygosity across the genome, were assessed by employing the analytic pipeline described in S8 Fig (see methods for details). To determine ploidy, read depth was assessed in conjunction with flow cytometry data (Figs 4, and S5 and S9).

Hybrid progeny from crosses between wild-type *C. neoformans* and *C. deneoformans* parental strains, as well as those from unilateral and bilateral *msh2Δ* hybrid crosses, displayed high

rates of aneuploidy (Figs 4, S9 and S10 and S4 Table). We characterized the number of instances in which each chromosome was aneuploid by determining which chromosomes had been gained or lost relative to the euploidy estimated by FACS analysis (i.e. variations from 1*n* or 2*n*). Aneuploidies involving all chromosomes were observed, with the exception of two homologous chromosome pairs: *C. neoformans* Chr5/*C. deneoformans* Chr4 and *C. neoformans* Chr 6/*C. deneoformans* Chr5 (S10A and S10B Fig and S4 Table). All hybrid progeny had three or fewer aneuploid chromosomes, with the exception of a progeny from a wild-type cross, YX6, which was disomic for five chromosomes. We also observed a trend in which larger chromosomes were less likely to be aneuploid, although there was not a significant correlation between chromosome length and likelihood of aneuploidy (S10C Fig and S4 Table).

The majority (16/27) of the hybrid progeny were diploid (2*n*, 12/27 progeny) or nearly diploid (2*n*+1 or 2*n*-1, 4/27 progeny), and all progeny derived from unilateral and bilateral *msh2Δ* crosses were close to diploid (S4 Table). Interestingly, hybrid progeny from bilateral *msh2Δ* mutant crosses were nearly completely heterozygous across a majority of the genome and displayed fewer LOH events (Figs 4, S9 and S11 and S5 Table). Quantifying this heterozygosity, hybrid progeny derived from bilateral *msh2Δ* x *msh2Δ* mutant crosses had significantly more heterozygosity ($p<0.05$, Kruskal-Wallis test, Dunn's test), with a genome-wide average of 96% heterozygosity relative to hybrid progeny derived from wild-type H99α x JEC20**a** crosses, which had an average of 67% heterozygosity across their genomes (S11 Fig and S5 Table).

In contrast to the results from the hybrid progeny, progeny derived from intraspecific wild-type, unilateral *msh2Δ*, and bilateral *msh2Δ* *C. neoformans* and *C. deneoformans* crosses were largely haploid based on the combination of FACS data and WGS (S6 and S12 Figs). Out of the 38 intraspecific progeny for which WGS was obtained, only three progeny from *C. neoformans* intraspecific crosses were not haploid: progeny 5 from a unilateral KN99α *msh2Δ* x KN99**a** cross, which was estimated to be near diploid (2*n*-1; missing one copy of Chr13), and progeny 3 and 4 from a bilateral KN99α *msh2Δ* x KN99**a** *msh2Δ* cross (both 1*n*+1; gained one copy of Chr2 and Chr7, respectively) (S12 Fig). All progeny from all *C. deneoformans* intraspecific crosses were haploid by both FACS analysis and WGS (S6 and S12 Figs).

### Hybrid progeny from *msh2Δ* genetic crosses do not exhibit increased meiotic recombination frequencies but do show instances of *de novo* telomere addition

Whole-genome sequencing revealed instances of potential meiotic recombination in the hybrid progeny derived from the wild-type, unilateral *msh2Δ*, and bilateral *msh2Δ* crosses. Due to the high levels of heterozygosity across the genomes of many *C. neoformans* x *C. deneoformans* hybrid progeny, and because all progeny were recovered through random spore dissection, we were not able to detect heteroduplex DNA, an indicator of recombination in MMR-deficient strains. Additionally, it is possible that some recombination events might not be detected by merely assessing the depth of reads aligning to each parental reference strain. For instance, if a hybrid progeny inherited both homeologous chromosomes involved in a meiotic reciprocal recombination event, read depth coverage would be equal across both parental chromosomes. To ensure that additional recombination events like these were not being missed, Illumina paired-end reads were aligned to a combined reference that included both parental genomes to identify additional possible recombination sites. This analysis was based on the assumption that if a recombination event occurred, the forward and reverse reads of a pair would align to different chromosomes. Different filtering thresholds were applied based on mapping quality and the number of read-pairs supporting the event. The results of these analyses, using more or less stringent filtering thresholds to detect recombination events,

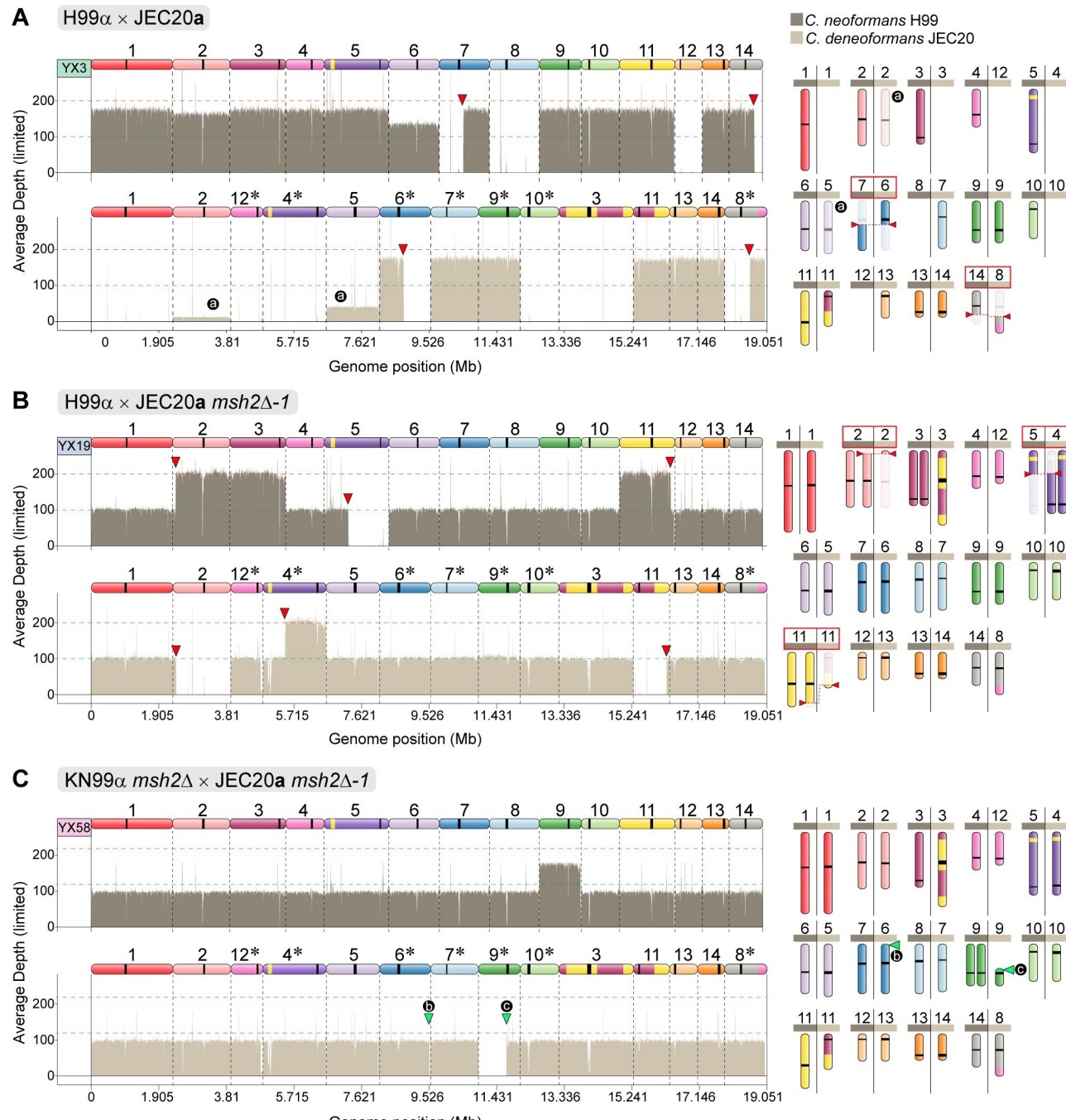

**Fig 4. Nuclear genome composition of *C. neoformans* x *C. deneoformans* hybrid progeny reveals substantial aneuploidy and recombination.** Representative sequencing read-depth coverage and inheritance patterns of progeny derived from **(A)** a wild-type H99α x JEC20**a** genetic cross, **(B)** a unilateral H99α x JEC20**a** *msh2Δ* genetic cross, and **(C)** a bilateral KN99α *msh2Δ* x JEC20**a** *msh2Δ* genetic cross. For each progeny, sequencing coverage plots (normalized to the genome-wide average coverage) are colored according to each parental species contribution as shown in the key on the top right, and a schematic representation of the inferred karyotype is depicted on the right. Homeologous chromosomes are color coded based on the H99 reference and asterisks in JEC21 indicate chromosomes in reverse-complement orientation (See S8 Fig for details). Red arrowheads mark recombination breakpoints between homeologous chromosomes and/or loss of heterozygosity (also highlighted by red boxes in the karyotype panels). Circular black labels: (a) indicate changes in ploidy in a subset of the population of cells that were sequenced; (b) and (c) mark, respectively, chromosome breaks at a transposable element near the end of the *C. deneoformans* Chr6 and at the centromere of Chr9, which were both repaired by *de novo* telomere addition (green arrowheads). Note the chromosome order of each parent; JEC21α contigs have been reordered to maximize collinearity with the H99α contigs.

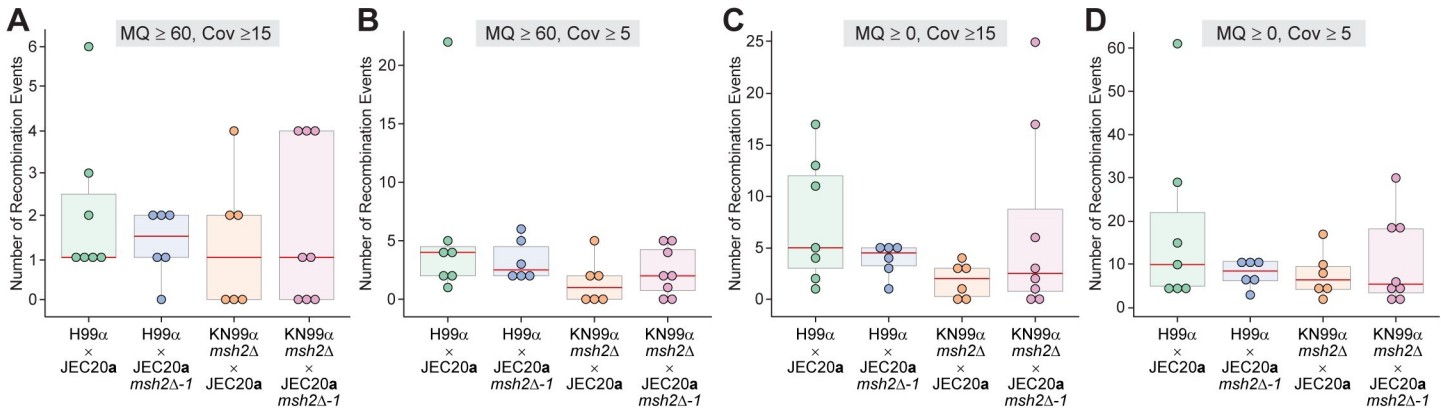

**Fig 5. Distribution of the number of recombination events detected in the hybrid progeny based on read-pairs aligning to different chromosomes.** Four different filtering thresholds were applied to detect potential instances of recombination across the genomes of 27 *C. neoformans* x *C. deneoformans* hybrid progeny: **(A)** Mapping quality (MQ) = 60 and at least 15 read-pairs (Cov) supporting each event; **(B)** MQ = 60 and at least 5 read-pairs supporting each event; **(C)** MQ > = 0 and at least 15 read-pairs supporting each event; and **(D)** MQ > = 0 and at least 5 read-pairs supporting each event. Each point represents the number of recombination events detected across the whole genome of a single hybrid progeny. In the box and whisker plots, red lines represent the median, shaded boxes represent the interquartile ranges (IQRs), upper and lower whiskers show the largest or smallest observations, respectively, that lie within 1.5 * IQR of the upper and lower quartiles, respectively. Outliers are included.

show similar frequencies of recombination events for each of the different types of hybrid crosses (Fig 5 and S6 Table). With the strictest filtering thresholds (at least 15 read-pairs supporting the event and a mapping quality of 60), a median of 1 recombination event was detected across the whole genomes of progeny derived from wild-type, unilateral KN99α *msh2Δ* x JEC20**a**, and bilateral *msh2Δ* crosses, while a median of 1.5 events were detected in progeny from unilateral H99α x JEC20**a** *msh2Δ-1* crosses. Furthermore, with the least strict thresholds (at least 5 read-pairs supporting and mapping quality greater than or equal to 0), a median of 10 recombination events were detected in progeny from a wild-type cross, a median of 8.5 and 6.5 events were detected in progeny from both unilateral crosses (H99α x JEC20**a** *msh2Δ-1* and KN99α *msh2Δ* x JEC20**a**, respectively), and a median of 5.5 events were detected in progeny from bilateral crosses (Fig 5 and S6 Table). These results showed no increase in meiotic recombination in hybrid progeny derived from parents lacking Msh2.

High-confidence recombination sites were mapped onto the H99 reference genome along with the distribution of SNPs differing between H99α and JEC20**a**, as well as the distribution of repetitive elements (Fig 6A). One might expect that recombination events in hybrid progeny would occur at regions with higher homology between the two parental genomes (i.e. regions with lower SNP densities). However, analysis of the SNP densities within 1 kb on either side of each recombination site showed that high-confidence recombination events did not occur in regions with significantly lower or higher SNP densities (Fig 6B). Moreover, no high-confidence recombination events were detected within the ~40-kb region that shares 98.5% identity between the two parents, known as the identity island [27]. Additionally, no tracts ≥ 300 nucleotides in length with complete homology between the two parental genomes were identified, including within the identity island, suggesting that recombination events were not being missed due to limitations of the sequencing methods used. It also did not appear that recombination events were more likely to occur at repetitive elements (Fig 6A). Interestingly, one of the recombination events was near the mating-type locus, a known hotspot for recombination in *Cryptococcus* [57].

Another unexpected finding was a phenomenon associated with hybrid progeny derived from unilateral or bilateral *msh2Δ* crosses in which chromosome breaks occurred at centromeres, transposable elements, and other regions in the genome and appeared to have been

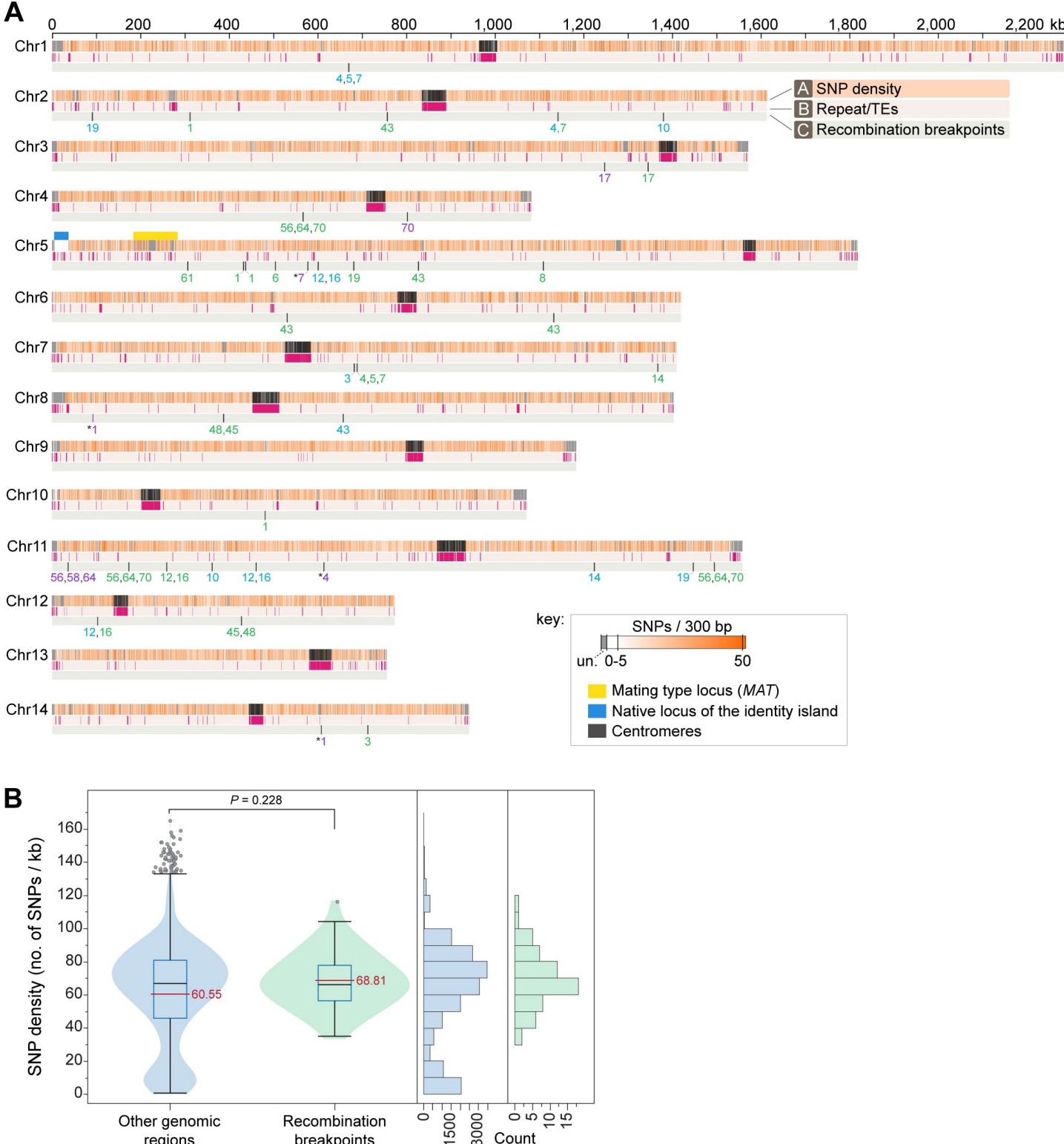

**Fig 6. Recombination breakpoints identified in the hybrid progeny are not associated with repeat-rich regions nor with lower SNP density regions between the two species. (A)** Plot showing the distribution of *C. deneoformans* SNPs on the 14 chromosomes of *C. neoformans* H99 (reference), repeat content (repeats and transposable elements identified by RepeatMasker), and the location of high-confidence recombination breakpoints identified in the hybrid progeny. SNPs were calculated in 300-bp windows and plotted as a heatmap color-coded as given in the key. Regions depicted in grey represent highly divergent regions between the two reference strains and are indicated in the key as unalignable (un.). The only more closely related region (~98.5% sequence similarity) shared between the two species is indicated by a blue bar and

corresponds to an ~40 kb region that resulted from a nonreciprocal transfer event (introgression) from *C. neoformans* to *C. deneoformans* [27]. The numbers below each recombination breakpoint correspond to the YX hybrid progeny strains (the YX prefix was omitted for simplicity of visualization) and are color-coded as: purple, when supported by MQ60-Cov15 and MQ60-Cov5; blue, when supported by MQ60-Cov15 and read depth; and green, when supported by MQ60-Cov15, MQ60-Cov5 and read depth (see methods for details). Some of the recombination breakpoints seem to be associated with recombination events between non-homeologous chromosomes of the two species and are marked with asterisks. **(B)** Violin plot, boxplots, and frequency histograms showing the SNP density within 1kb regions surrounding the high-confidence recombination breakpoints (green) compared to other genomic regions (blue). Red line, black line, blue box, and grey circles denote the mean value, median value, interquartile range, and outliers, respectively. The SNP density in the recombination breakpoint-containing regions is not statistically significantly different from the rest of the genome (Mann-Whitney test).

repaired by *de novo* addition of telomeric repeats (Figs 4C, S9 and S13 and S7 Table). Of the 15 instances of *de novo* telomere addition in the hybrid progeny, 5 occurred at repetitive elements, such as centromeres and transposons. We also assessed whether or not this phenomenon occurred in any of the progeny derived from intraspecific crosses, with the hypothesis that this type of chromosomal breakage and *de novo* telomere addition would only be viable in the context of a diploid, such that no essential genes are lost if only one homolog breaks. Accordingly, in the intraspecific progeny, we only identified one instance of *de novo* telomere addition at the end of chromosome 13 in progeny 3 from a bilateral KN99α *msh2Δ* x KN99**a** *msh2Δ* cross (S14 Fig and S7 Table). This event occurred ~17 kb from the end of the chromosome and only resulted in the loss of three genes encoding hypothetical proteins (CNAG_07919, CNAG_06256, and CNAG_07920) as well as a gene encoding a transporter belonging to the major facilitator superfamily (MFS) (CNAG_06259) (S14 Fig), indicating these are not essential genes.

## Discussion

In this study, we investigated the role of MMR in maintaining the species boundary between two closely related and prevalent opportunistic human fungal pathogens, *C. neoformans* and *C. deneoformans*. The MMR pathway is known to play a highly conserved role in blocking recombination between homeologous DNA sequences during meiosis, serving as a post-zygotic barrier in addition to its role in DNA repair. Findings from previous studies in prokaryotic and eukaryotic models indicated that inactivating the MMR pathway by genetically deleting pathway components, particularly Msh2, allows increased homeologous recombination during meiosis [8,10,11]. Therefore, we hypothesized that lack of Msh2 in *Cryptococcus* would allow increased pairing and recombination between homeologous chromosomes during meiosis, leading to decreased rates of chromosome nondisjunction, and ultimately the production of more viable progeny with fewer instances of aneuploidy.

Fluctuation analysis and characterization of the mutational spectra of *C. deneoformans* *msh2Δ* mutants confirmed that Msh2 plays a similar role in DNA MMR in *C. deneoformans* as has been previously observed across eukaryotes and in other pathogenic *Cryptococcus* species [49,50]. Similar to results from previous studies in prokaryotic and eukaryotic microorganisms, germination frequencies of *C. neoformans* x *C. deneoformans* hybrid progeny derived from bilateral *msh2Δ* crosses increased significantly relative to wild-type hybrid germination frequencies [8,10,11]. This was in contrast to the reduced viability observed in bilateral *msh2Δ* intraspecific crosses, which decreased by 12% compared to wild-type *C. neoformans* intraspecific crosses and 42% compared to wild-type *C. deneoformans* intraspecific crosses; this reduced viability is likely due to the hypermutator phenotype of *msh2Δ* mutants, as has been observed in previous studies in *S. cerevisiae* [10,54].

Hybrid progeny derived from genetic crosses between *C. neoformans* and *C. deneoformans* isolates are known to display unique phenotypes. For example, hybrids, especially those generated under laboratory conditions, can exhibit self-fertility, producing hyphae and in some

cases basidia and spores under mating-inducing conditions [41,45]. The majority of the hybrid progeny derived from the crosses described here displayed phenotypes in accord with previously published findings for *C. neoformans* x *C. deneoformans* hybrids [41,43]. However, a novel transgressive phenotype was exhibited by many hybrid progeny: the ability to produce hyphae and in some cases, basidia and sparse basidiospores, on nutrient-rich YPD medium in sealed plates in the light. This was surprising, because environmental cues known to suppress mating in *Cryptococcus* include nutrient-rich environments, such as YPD medium, as well as light and high levels of humidity and carbon dioxide [33,36,58–60]. The formation of sexual reproduction structures by the hybrid progeny under nutrient-rich, light, sealed conditions represents sexual reproduction of *Cryptococcus* in a novel environment, which has never been shown to induce filamentation for any *C. neoformans* or *C. deneoformans* strain previously. The isolation of self-fertile progeny provides an important example of how hybridization can enable *Cryptococcus* to access a dimorphic state under conditions that are not conducive for the parental isolates to sexually reproduce. The ability to produce spores, the infectious propagules of *Cryptococcus* human pathogens, in a previously prohibitive environment also provides the opportunity to produce more infectious propagules, which could potentially result in a higher rate of infection. This is in line with previous studies which found that up to 30% of clinical isolates in Europe were *C. neoformans* x *C. deneoformans* hybrids [38–40]. These results, along with recent studies that identified Msh2-deficient *C. neoformans* clinical isolates [50], suggest that certain *Cryptococcus* hybrids could potentially contribute to the emergence of a new pathogen, similar to the recent emergence of a wheat stem rust pathogen lineage that is the result of a hybridization with no subsequent recombination or chromosomal reassortment [61].

Unlike their haploid parents, *C. neoformans* x *C. deneoformans* hybrid progeny typically have relatively unstable aneuploid or diploid karyotypes [43–45]. All hybrid progeny derived from *C. neoformans* x *C. deneoformans* crosses in this study were aneuploid based on both FACS analysis and whole-genome sequencing data. Three hybrid progeny isolated from wild-type *C. neoformans* x *C. deneoformans* crosses were close to haploid and all other hybrid progeny derived were diploid or nearly diploid. Additionally, all hybrid progeny were euploid or nearly euploid, with three or fewer aneuploid chromosomes, with only one exception. These findings are similar to those from Parry and Cox, who found that *S. cerevisiae* progeny dissected from a triploid were close to euploid, suggesting only a limited number of aneuploid chromosomes are tolerated [62]. We also observed instances where the read-depth of several aneuploid chromosomes was lower than expected, which likely represent events of chromosome loss among a fraction of the cells that were sequenced at the whole-genome level, reflecting the karyotypic instability of these progeny.

The higher ploidy levels in progeny derived from crosses involving *msh2Δ* mutants were unexpected because MMR mutants in yeast produce progeny with lower levels of aneuploidy [10]. It is possible that the diploid or near diploid *C. neoformans* x *C. deneoformans* progeny were the meiotic products of a tetraploid, generated by the fusion of two diploids that formed within the mating patch. However, interspecific progeny derived from divergent *Saccharomyces* tetraploids display much higher germination frequencies (>90%), which is very different from the frequencies associated with these *C. neoformans* x *C. deneoformans* hybrid progeny, making it unlikely that they are derived from tetraploids [63]. It is also possible that the high frequency of diploid progeny could be indicative of a failed meiosis. However, *Cryptococcus* mutants lacking known meiotic genes, such as those encoding the meiosis-specific endonuclease Spo11 or the key meiotic regulator Dmc1, are either unable to produce viable progeny or unable to efficiently produce basidiospores, respectively [64,65]. In contrast to the abnormal sexual reproduction structures produced by these meiosis-deficient mutants, the basidia

produced by hybrid crosses in this study generated four lengthy spore chains. On the other hand, it is possible that without Msh2 during the hybrid meiosis, initiation of recombination between homeologous sequences will not be prevented effectively, and meiosis will therefore proceed to a degree while homeologous chromosomes are linked by strand invasions, increasing the number of viable progeny. This could also lead to increased chromosomal nondisjunction in meiosis I or potentially a skipping of the reductional meiotic division if homeologous chromosomes remain linked by strand invasions, which could explain the near diploid genomes of hybrid progeny from the unilateral and bilateral *msh2Δ* crosses. The strand invasions between homeologous chromosomes might be resolved as non-crossovers or crossovers, although the crossovers may be inviable, which could also explain the increased number of basidia lacking spore chains in the bilateral *msh2Δ* hybrid matings. The higher ploidy in hybrid progeny from bilateral *msh2Δ* x *msh2Δ* crosses was also associated with significantly more heterozygosity across their genomes than their counterparts from wild-type *C. neoformans* x *C. deneoformans* crosses. It is possible that this higher level of heterozygosity contributes to the increased viability by masking deleterious alleles or overcoming Bateson-Dobzhansky-Muller genetic incompatibilities, as has been observed in previous studies of *C. neoformans* x *C. deneoformans* hybrids [48,66].

An interesting phenomenon associated with progeny derived from unilateral and bilateral hybrid *msh2Δ* crosses as well as a single intraspecific progeny from a bilateral *msh2Δ* cross was the *de novo* addition of telomeric repeat sequences to various locations in the genome including centromeric repeat sequences, non-centromeric transposable elements, and the rDNA repeat locus following chromosomal breaks. Although the addition of telomeric repeats at non-telomeric sites can promote the stabilization of a broken chromosome, it also often leads to the loss of a large portion of a chromosome, which would normally threaten cell viability. However, the presence of both homeologous chromosomes from each parental species in these hybrid progeny alleviates this problem. In the haploid intraspecific progeny, only one instance of *de novo* telomere addition was observed, and in this case only a small portion of the chromosome (~17kb) was lost, thus avoiding the deleterious effects that might be associated with the larger losses observed in the hybrid progeny if they were to happen in a haploid background. Loss of Msh2 has been shown to promote telomeric recombination in yeast, and our results suggest Msh2 might be mediating a similar anti-recombination mechanism at telomeric and other repetitive loci in *Cryptococcus* [67]. Further supporting this, *de novo* telomeric repeat addition has been observed in *Cryptococcus* following CRISPR-mediated double-stranded breaks at centromeres [68]. Overall, our results suggest that in *Cryptococcus* crosses involving *msh2Δ* mutants, chromosomes may be more prone to double-stranded breaks, or that the normally occurring double-stranded breaks are unable to be properly repaired, and loss of Msh2 promotes *de novo* telomere addition at these sites.

Many instances of LOH and recombination were observed in the hybrid progeny assessed in this study. One caveat to note regarding the recombination events identified through WGS, is that *C. neoformans* x *C. deneoformans* hybrids have been known to experience loss of heterozygosity during mitotic growth [45], and results from other previous studies also suggest that mitotic recombination can occur during mating itself [47,69]. Unexpectedly, instances of recombination were detected in progeny derived from each of the different types of hybrid crosses (wild-type, unilateral, and bilateral *msh2Δ*), but no increase in meiotic recombination was observed in hybrid progeny derived from parents lacking Msh2. Although *Cryptococcus* has lower frequencies of meiotic recombination compared to *S. cerevisiae* – approximately 1.27 crossovers per chromosome per progeny derived from intraspecific bisexual crosses [70], and this frequency is estimated to decrease by six- to seven-fold in *C. neoformans* x *C. deneoformans* hybrid progeny [46] – we expected to see significantly higher rates of recombination

in hybrid progeny from *msh2Δ* crosses compared to wild-type crosses based on previous studies in both prokaryotes and eukaryotes [8,10,11]. Based on read-depth analysis as well as detecting recombination events by aligning whole-genome sequencing from the progeny to a combined reference genome with both parental species, no increase in recombination was observed in crosses involving either a single parent lacking Msh2 or both parents lacking Msh2. One potential explanation for this observation could be that the level of sequence divergence between *C. deneoformans* and *C. neoformans* is large enough that meiotic recombination will be inefficient, even in the absence of MMR. However, previous studies on meiotic recombination in *S. cerevisiae* indicate that although recombination is less efficient at high levels of sequence divergence, loss of MMR leads to approximately a 24-fold increase in meiotic recombination between 15% divergent sequences, the same level of divergence as between *C. neoformans* and *C. deneoformans* [27,29,71]. Furthermore, studies on mitotic recombination in *S. cerevisiae* also similarly indicate that even at 26% sequence divergence, loss of MMR leads to a 55-fold increase in mitotic recombination [72,73]. Conversely, studies in other models have observed instances where loss of Msh2 did not lead to increased recombination frequencies between substrates with as little as 1% sequence divergence, up to 25% divergence [74–76].

In summary, this study illustrates several key findings on the roles of MMR in *Cryptococcus*. In *C. deneoformans*, *msh2Δ* mutants behave as hypermutators on various selective media and acquire INDELs in homopolymeric nucleotide runs. Hybrid *C. neoformans* x *C. deneoformans* progeny dissected from genetic crosses involving *msh2Δ* mutants generally displayed increased germination frequencies compared to those from wild-type crosses. Hybrid progeny derived from *msh2Δ* crosses displayed phenotypes and karyotypes characteristic of *C. neoformans* x *C. deneoformans* hybrid strains, such as diploidy/aneuploidy and self-fertility, and some progeny displayed a novel, transgressive phenotype in which they were capable of producing hyphae, basidia, and basidiospores on a glucose- and nutrient-rich medium in the light. The increased viability of hybrid progeny derived from bilateral *msh2Δ* crosses suggests that loss of Msh2 in *Cryptococcus* may allow homeologous chromosomes to pair more efficiently during meiosis. However, the observation that loss of Msh2 did not seem to increase the frequency of meiotic recombination between *C. neoformans* and *C. deneoformans* homeologous chromosomes was highly unexpected based on many previous studies, particularly those in yeast. These results suggest that Msh2 plays a role in maintaining the species boundary between *C. neoformans* and *C. deneoformans*, albeit an unexpected one. Additionally, these results suggest alternative pathways or additional MMR components may play different or more important roles in maintaining species boundaries in *Cryptococcus* than in other previously studied organisms; proteins like the DNA helicases Mph1 and Sgs1, which have been shown to block homeologous recombination and play significant roles in chromosome nondisjunction in budding yeast, would be ideal candidates for further investigation [77–79]. Thus, future studies identifying the robust post-zygotic mechanisms that ultimately maintain integrity by blocking homeologous recombination between these two closely related species will be of great interest.

## Materials and methods

### Strains and growth

The *C. neoformans* and *C. deneoformans* strains described in this study are listed in S8 Table. Strains were stored at -80°C in liquid yeast extract peptone dextrose (YPD) supplemented with 15% glycerol. Strains were inoculated on YPD agar plates, initially grown at 30°C for 3 days, and then maintained at 4°C. Due to the hypermutator phenotypes associated with *msh2Δ* strains and the genomic instability associated with *C. neoformans* x *C. deneoformans* hybrids, strains used in the experiments of this study were not maintained for more than two weeks at

4˚C on YPD agar plates and fresh cells from frozen glycerol stocks were inoculated to new YPD agar plates at the end of each two-week period.

### Generation of *msh2Δ* deletion mutants

The open-reading frame of the *MSH2* gene (gene ID: CNA07480) in the *C. deneoformans* JEC20**a** [80] genetic background was replaced with the gene encoding the dominant drug-resistance marker for nourseothricin resistance, *NAT*, by homologous recombination via bio-listic transformation as previously described [81]. Following transformation and selection on YPD + 100 μg/mL nourseothricin agar medium, genomic DNA was isolated from candidate mutants with the MasterPure DNA purification kit (Epicentre) and PCR followed by gel electrophoresis confirmed correct integration of the *NAT* dominant resistance marker. The locations of the primers used to generate the deletion allele and confirm deletion mutants are depicted in S1A Fig and their sequences are given in S9 Table. To generate congenic strains of opposite mating types for the intraspecific *C. neoformans* and *C. deneoformans msh2Δ* crosses, KN99**a** [82] was crossed with the KN99α *msh2Δ* mutant and JEC20**a** *msh2Δ-1* was crossed with JEC21α [80], respectively. Progeny were isolated and PCR was used to confirm that they inherited the *msh2Δ* deletion construct at the endogenous *MSH2* locus as well as the appropriate mating type, and did not inherit a functional *MSH2* allele.

### Fluctuation analysis to quantify mutation rates

Fluctuation analysis was utilized to quantify the mutation rates, or the number of mutations per cell per generation, of the JEC20**a** *msh2Δ* mutants. The wild-type strain JEC20**a** served as a negative control and the *msh2Δ* mutant from the 2015 Madhani deletion collection in the KN99α genetic background served as a positive control [51]. For each strain, including controls, ten 5 mL YPD liquid cultures were each inoculated with a single colony from a YPD agar stock plate. Cultures were incubated overnight at 30˚C. After incubation, cultures were pelleted at 3,000 x *g*, washed twice with 5 mL dH$_2$O, and resuspended in 4 mL dH$_2$O. 100 μL of undiluted, washed cells was plated directly to YPD + 100 ng/mL rapamycin + 1 μg/mL FK506 or yeast nitrogen base (YNB) + 1 mg/mL 5-fluoroorotic acid (5-FOA) solid agar medium. Washed cells were diluted 1:100,000 and 100 μL of the dilution was plated to YPD solid agar medium. Inoculated YPD and YNB+5-FOA plates were incubated at 30˚C for 4 or 14 days, respectively. Inoculated YPD+rapamycin+FK506 plates were incubated at 37˚C for 14 days. Following incubation, colonies on each of the media were counted and mutation frequencies were calculated with the FluCalc program, which utilizes the Ma-Sandri-Sarkar maximum-likelihood estimation (MSS-MLE) equation for calculations [83].

### Mutation spectra analysis

Single resistant colonies from fluctuation analyses were streak purified to YPD + rapamycin + FK506 medium and grown for 3 days at 37˚C. Genomic DNA was isolated using the Master-Pure Yeast DNA Purification Kit (Epicenter Biotechnologies, Madison, WI), and the *FRR1* gene was PCR-amplified with Phusion High-Fidelity DNA Polymerase (NEB, Ipswich MA, USA). PCR products were subjected to gel electrophoresis, extracted using a QIAgen gel extraction kit, and mutations were identified through classical Sanger sequencing at Genewiz. Fisher's exact probability test was used to calculate statistically significant differences between the frequencies of 1-bp INDEL mutations compared to other types of mutations in YPD + rapamycin + FK506-resistant colonies from strains lacking *MSH2* compared to the wild-type JEC20**a** strain using the VassarStats online software (http://vassarstats.net).

## Papillation assays

Papillation assays were conducted on YNB solid agar medium supplemented with 1 mg/mL 5-FOA, 100 µg/mL 5-fluorocytosine (5FC), or 100 µg/mL 5-fluorouridine (5FU). For this assay, ten independent YPD liquid cultures were inoculated with ten single colonies from YPD agar plates and incubated overnight at standard laboratory conditions. Following overnight culture, cells were pelleted at 3,000 x *g* and resuspended in 2 mL dH$_2$O. Sterile cotton swabs were then used to inoculate quadrants of the agar plates with each independent overnight culture. The inoculated agar plates were incubated at 30˚C for 6 days to allow sufficient growth to visualize resistant colonies.

## Genetic crosses and progeny dissection

All genetic crosses were conducted on Murashige and Skoog (MS) agar medium following Basic Protocol 1 for mating assays as previously described [84]. The frequency of bald basidia was calculated by imaging random areas surrounding two mating patches per cross, counting the number of bald basidia and basidia producing basidiospores, and the frequencies from the two mating patches were averaged. For each genetic cross, two independent mating patches were assessed; for each mating patch, over 130 total basidia were assessed across at least 11 images. Images used to quantify bald basidia were taken on a Zeiss Axio Scope.A1 with camera. Random basidiospore dissection was performed as described in Basic Protocol 2 [84]. Following dissection, the micromanipulated basidiospores were germinated for up to 3 weeks on YPD agar plates sealed with parafilm and incubated on the laboratory benchtop. Images of self-fertile hybrid progeny derived from bilateral *msh2Δ* crosses producing sexual structures on YPD agar plates following dissection were taken with an Accu-Scope EXC-500 microscope with an attached Nikon DXM1200F microscope camera.

## Phenotyping and genotyping of hybrid progeny

Primers designed to specifically amplify only the *C. neoformans* STE20**a**, *C. neoformans* STE20α, *C. deneoformans* STE20**a**, or *C. deneoformans* STE20α alleles aided in identifying the mating types of the hybrid progeny (primers listed in S9 Table). To assess self-filamentation, hybrid progeny were spotted onto MS agar plates and incubated for 14 days at room temperature (approximately 24˚C) in the dark as described in Basic Protocol 1 [84]. Filamentation was assessed via microscopy after 14 days of incubation with an Accu-Scope EXC-500 microscope with an attached Nikon DXM1200F microscope camera.

## Flow cytometry

Cells were patched onto YPD agar medium and incubated at 30˚C overnight; strains that exhibited slow growth were incubated at 30˚C for three days. Cells were harvested by scraping a 2 mm sized colony with a toothpick and resuspending in 1 mL PBS buffer. Cells were washed once with 1 mL PBS and then fixed in 1 mL 70% ethanol overnight at 4˚C. After fixation, cells were washed once with 1 mL NS buffer (10 mM Tris-HCl pH = 7.6, 150 mM sucrose, 1 mM EDTA pH = 8.0, 1 mM MgCl$_2$, 0.1 mM ZnCl$_2$, 0.4 mM phenylmethylsulfonyl fluoride, 1 mM β-mercaptoethanol) and resuspended in 180 µl NS buffer supplemented with 5 µl propidium iodide (0.5 mg/mL) and 20 µl RNase A (10 mg/mL). Cells were incubated covered, overnight at 4˚C with shaking. Prior to analysis, 50 µl of cells were diluted in 2 mL of 50 mM Tris-HCl (pH = 8.0) in a 5 mL falcon tube. Flow cytometry was performed on 10,000 cells and analyzed on the FL1 channel on a Becton-Dickinson FACScan.

## Whole-genome sequencing

Single colonies of strains for whole-genome sequencing were inoculated into 50 mL of liquid YPD and grown overnight at 30°C in standard laboratory conditions. Overnight cultures were then pelleted at 3,000 x *g* in a tabletop centrifuge and subsequently lyophilized. High molecular weight DNA was extracted from lyophilized cells following the CTAB protocol as previously described [85]. Genomic DNA libraries were constructed with a Kapa HyperPlus library kit for 300 bp inserts and sequenced at the Duke Sequencing and Genomic Technologies Shared Research core facility. Libraries were sequenced using paired-end, 2 x 150 bp reads on an Illumina HiSeq 4000 platform. The BioProject accession numbers for each sample are provided in S10 Table.

## Whole-genome and chromosome composition analyses of the hybrid progeny

*C. deneoformans* strain JEC21α is the congenic mating partner of JEC20**a**, which was obtained in an earlier study by selecting *MAT*α progeny after ten rounds of backcrossing to JEC20**a** [80]. The genomes of *C. deneoformans* JEC21α and JEC20**a** strains are therefore nearly identical, with the exception of 5,322 SNPs mainly distributed over three genomic regions (including the mating-type locus) [86]. Because a highly contiguous *de novo* genome assembly of JEC20**a** is not currently available, the genome assembly of strain JEC21α (GCA_000091045.1) was used for all comparisons. The nuclear genomes of the two parental strains, *C. neoformans* H99α and *C. deneoformans* JEC21α, were compared by performing whole-genome alignments with Satsuma (https://github.com/bioinfologics/satsuma2) [87], using default parameters. The output of Satsuma was input to the visualization tools "BlockDisplaySatsuma" and "ChromosomePaint", included in the same package to generate a postscript file. For representation purposes, chromosome color codes were modified in Adobe Illustrator, and centromeres and other genomic features (rDNA and *MAT* loci) were superimposed at scale based on their respective genomic coordinates. Characterization of the chromosome composition of the hybrid progeny followed the procedure summarized in S8A Fig. The combined nuclear reference genome used in this study was built with the genome assemblies of the two parental strains after reordering and reorienting the JEC21α contigs to maximize collinearity with the H99α assembly (S8B Fig), using the Mauve Contig Mover tool [88]. A dot plot analysis comparing the H99α assembly with the JEC21α rearranged assembly (S8B Fig) was performed with D-Genies application [89], which uses minimap2 [90] for aligning the two genomes. Raw Illumina paired-end reads of the selected progeny and the combined reference genomes were input into the sppIDer pipeline [91], which is a wrapper that sequentially maps the Illumina short reads to the combined reference, performs quality filtering (MQ > 3), and generates depth of coverage plots (Figs 4 and S9). For each progeny, the number of chromosomes and the ploidy were estimated from the sppIDer plots in conjunction with the flow-cytometry data. Chromosomal aberrations, e.g. due to recombination, or chromosome breakage followed by *de novo* telomere addition, were inferred from the sppIDer plots, and further validated by visual inspection of the mapped reads in IGV [92].

## Heterozygosity of the hybrid progeny

To inspect the heterozygosity levels of the hybrid progeny, the same set of Illumina paired-end reads were mapped to the *C. neoformans* H99α reference genome, using the BWA-MEM short-read aligner (v0.7.17-r1188) with default settings [93]. SNP discovery, variant evaluation, and further refinements were carried out with the Genome Analysis Toolkit (GATK) best-

practices pipeline [94,95] (v4.0.1.2), including the use of Picard tools to convert SAM to sorted BAM files, fix read groups (module: 'AddOrReplaceReadGroups'; SORT_ORDER = coordinate), and mark duplicates. Variant sites were identified with HaplotypeCaller from GATK and only high-confidence variants that passed filtration were retained (the "VariantFiltration" module used the following criteria: DP < 20 || QD < 2.0 || FS > 60.0 || MQ < 40.0 || SOR > 4.0). The genome-wide level of heterozygosity was defined as the ratio of the number of heterozygous SNPs divided by the total number of SNPs (i.e. heterozygous and non-reference homozygous SNPs), and was calculated from the resulting VCF files on a per-individual basis after extracting the corresponding sites using the module VariantsToTable from GATK. Due to the high variance in heterogeneity across the genomes of progeny from the different genetic crosses and unequal numbers of progeny analyzed from each different genetic cross, the non-parametric Kruskal-Wallis test followed by Dunn's test were performed using JMP v15 (SAS Institute).

### Detection of variants in the genome of the JEC20a *msh2Δ-1* mutant

To predict variants in the JEC20**a** *msh2Δ-1* mutant, we employed the variant calling procedure described above, using the JEC21 genome (GCA_000091045.1) as reference. The effect of a variant (SNPs and INDELs) was predicted and annotated using SnpEff [96]. Only variants of moderate and high impact were considered (i.e. excluding synonymous and non-coding variants), and variants in common between the mutant and the JEC20**a** parent were flagged as background mutations and excluded. Given the inability of the *msh2Δ-1* mutant to produce 5-FOA resistant colonies, we specifically focused on mutations in genes involved in the *de novo* and salvage pathways of pyrimidine biosynthesis. A single base-pair deletion was identified in the gene *FUR1*: deletion of a thymine at position 570229 on Chr5 (AE017345.1), GeneID: CNE02100 (AE017345.1:569,042–570,826). Mutations in components of the pyrimidine *de novo* biosynthesis pathway, such as *URA3*, have been demonstrated to be synthetically lethal with *FUR1* mutations in yeast [97,98]. Independent overnight cultures of each of the independent JEC20**a** *msh2Δ* mutants, a KN99α *fur1Δ* mutant, the JEC20**a** progenitor strain (negative control), and a KN99α *msh2Δ* mutant (positive control) were swabbed on YNB plates supplemented with 5-FOA to illustrate that both JEC20**a** *msh2Δ-1* and KN99α *fur1Δ* mutants are incapable of producing 5-FOA resistant colonies (S2B Fig). Furthermore, *fur1Δ* mutants are also known to be resistant to the antifungal drug 5-fluorouridine (5FU) and the clinically relevant antifungal drug 5-fluorocytosine (5FC) [52]. Papillation assays with the same strains used on 5-FOA were also conducted on YNB plates supplemented with 5FC or 5FU. As anticipated, the JEC20**a** *msh2Δ-1* and the KN99α *fur1Δ* mutant displayed congruent growth phenotypes and were resistant to both 5FC and 5FU, unlike the parental JEC20**a**, the KN99α *msh2Δ* strain, or any of the other JEC20**a** *msh2Δ* mutants (S2C and S2D Fig).

### Detection of recombination events based on read-pair alignment

To detect possible recombination events in *C. neoformans* x *C. deneoformans* hybrids, the respective reference genomes were retrieved from NCBI (accession: ASM301198v1 for H99 and ASM9104v1 for JEC21 as no genome assembly for JEC20 was available). As the H99 strain was not sequenced in this project, the respective Illumina paired-end reads were retrieved from SRA (SRR7042283). All Illumina paired-end libraries were filtered with the default parameters of Trimmomatic v0.36 [99]. Filtered reads of each strain (including the parental strains) were aligned on the combined reference (containing both H99 and JEC21 genome assemblies) with HaploTypo pipeline [100], using BWA-MEM v0.7015 [93], samtools v1.9 (Li 2011) and GATK v4.0.2.1 [95].

To detect recombination events, we assumed that at recombinant sites it would be possible to detect read-pairs with the two reads aligning to different chromosomes. Therefore, for each strain (including the parental strains), we obtained the coordinates of all the pairs with each read aligning to a chromosome of a different parental species. Only reads with at least 75 bp and a maximum of 3 mismatches were considered. As there were some differences between the parental JEC20 and the reference JEC21, variant calling was performed with HaploTypo pipeline [100], using GATK v4.0.2.1 [95]. Bedtools v2.26 [101] intersect was used to determine how many SNPs overlapped each read, and in cases in which more than 3 SNPs would overlap, the filter of mismatches was adjusted to that value. The final results were filtered based on mapping quality (MQ) and number of pairs supporting a given recombination event (Bedtools v2.26 [101] merge with a distance of 100bp). Four different analyses were performed using more or less stringent filters: i) MQ = 60 and at least 15 reads supporting the event (stricter); ii) MQ = 60 and at least five reads supporting the event; iii) MQ $> = 0$ and at least 15 reads supporting the event; iv) MQ $> = 0$ and at least five reads supporting the event (less strict). The script that automates this pipeline is available on GitHub (https://github.com/Gabaldonlab/detect_recombination/). As it was possible to detect recombination in the parental strains (false positives), Bedtools v2.26 [101] subtract was used to remove all the regions detected in hybrid progeny that were also detected in the parental strains.

## Aneuploidy and *de novo* telomere addition assessment in progeny from intraspecific crosses of *C. neoformans* and *C. deneoformans*

Paired-end reads of the progeny resulting from intraspecific crosses of *C. neoformans* H99α × KN99**a** and *C. deneoformans* JEC21α × JEC20**a**, or from intraspecific crosses involving their derived *msh2Δ* mutants, were mapped to the *C. neoformans* H99α and *C. deneoformans* JEC21α reference genomes, respectively, using the procedures described above. Gross aneuploidy of chromosomes was inferred from read counts collected in 1-kb non-overlapping intervals across the genome using the module "count_dna" from the Alfred package (v0.1.7) (https://github.com/tobiasrausch/alfred) and subjected to median normalization and log2 transformation. The resulting data was converted to binary tiled data (.tdf) using "igvtools toTDF" and plotted as a heatmap in IGV viewer. Chromosome breaks and *de novo* telomere addition were inferred by abrupt changes in read coverage within a chromosome and visually confirmed by read mapping in IGV.

## Testing the association between SNP density, repeat content, and the recombination breakpoints identified in the *C. neoformans* x *C. deneoformans* hybrid progeny

Recombination breakpoints in hybrid progeny could potentially occur at regions with higher sequence identity between the two parental genomes (i.e. regions with lower SNP densities). If no SNPs could be scored between the two species in genomic tracts smaller than 300 bp (corresponding to the read size) this would potentially lead to an underestimation of the recombination events. Therefore, to determine the distribution of SNPs differing between *C. neoformans* and *C. deneoformans*, 150-bp paired-reads of *C. deneoformans* JEC20**a** were mapped to a *C. neoformans* H99α reference genome using the methods described above. SNP density was calculated from the resulting VCF file in 300-bp bins using VCFtools with the option "—SNPdensity 300", parsed into a BED file format, and visualized as a density heatmap in IGV. Regions without mapped reads were identified by extracting intervals with no coverage using the output of "bedtools genomcov -bga". No regions of 300 bp were found without any SNP,

including the ~40-kb nearly identical region between the two parental genomes (~98.5% sequence identity) that was introgressed from *C. neoformans* to *C. deneoformans* [27].

Next, to examine whether high-confidence recombination events occur in regions with significantly lower or higher SNP densities, we compared the SNP densities within 1-kb on each side of each of the inferred recombination breakpoints, with other 1kb-binned genomic regions (excluding centromeres and the mating-type locus region), and plotted their density and distribution. Three sets of high-confidence recombination events were chosen for this analysis: (a) recombination events supported by mapping quality of 60 and by both 5 and 15 read-pairs (MQ60-Cov15 and MQ60-Cov5), but that could not be inferred directly from the read depth plots; (b) events supported by MQ60-Cov15 and the read depth plots, but not called when using MQ60-Cov5 as a filtering criteria; and (c) events supported by both MQ60-Cov15 and MQ60-Cov5 which were also directly inferred from the read depth plots.

Finally, to inspect if any of these recombination breakpoints were associated with genomic regions enriched in repeats or transposable elements, RepeatMasker (RepeatMasker Open-4.0 2013–2015; http://www.repeatmasker.org) was run with a library of previously characterized *C. neoformans* transposable elements [102] and *de novo*-identified repeat consensus sequences generated by RepeatModeler2 (https://github.com/Dfam-consortium/RepeatModeler). "Bedtools intersect" was employed to determine if any of the genomic regions associated with repeats or transposable elements overlapped with any of the recombination breakpoints.

### Aneuploidy of the hybrid progeny

Hybrid progeny were considered aneuploid if they were missing or had an extra chromosome compared to the theoretical expected number given their ploidy level as measured by FACS analyses, and irrespective of whether they inherited copies from each parent or from only one of the parents. For example, progeny YX4 was scored as euploid because it was determined to be 2*n* by FACS and has two copies of each chromosome, even though the two copies of chromosomes 4 and 11 were both inherited from *C. neoformans*. Another example, YX6, which by FACS is close to 1*n*, was scored as aneuploid for chromosomes 9, 11, 12, 13 and 14 (see S4 Table). The results of these analyses were plotted as the number of hybrid progeny with aneuploidies for each chromosome and genetic cross, and the total aneuploidy observed for all strains was plotted by chromosome length, which showed no correlation between chromosome size and aneuploidy.

### Supporting information

**S1 Table.** (A) Frequency of bald basidia produced by hybrid genetic crosses, and (B) one-way ANOVA and Tukey's HSD post hoc statistical tests for frequencies of bald basidia. (DOCX)

**S2 Table.** (A) Germination frequencies of progeny derived from hybrid and intraspecific crosses, and one-way ANOVA with Tukey's HSD post hoc statistical tests for germination frequencies of (B) *C. neoformans* x *C. deneoformans* hybrid progeny, (C) progeny from *C. neoformans* x *C. neoformans* intraspecific crosses, and (D) progeny from *C. deneoformans* x *C. deneoformans* intraspecific crosses. (DOCX)

**S3 Table.** (A) Self-filamentation on YPD medium in hybrid and intraspecific progeny, and (B) one-way ANOVA statistical test for self-filamentation of hybrid progeny. (DOCX)

**S4 Table. Aneuploidy assessment in the *C. neoformans* x *C. deneoformans* hybrid progeny.**
(DOCX)

**S5 Table.** (A) Heterozygosity levels of the hybrid progeny, and (B) Kruskal-Wallis and Dunn's statistical tests.
(DOCX)

**S6 Table. Recombination events detected in *C. neoformans* x *C. deneoformans* hybrid progeny.**
(XLSX)

**S7 Table. Genomic locations of telomeric repeats added *de novo* at chromosome breaks in progeny derived from unilateral and bilateral hybrid crosses of *C. neoformans* x *C. deneoformans msh2Δ* mutants, and in a single progeny derived from a bilateral intraspecific cross of *C. neoformans msh2Δ* mutants.**
(DOCX)

**S8 Table. Strains used in this study.**
(DOCX)

**S9 Table. Oligonucleotides used in this study.**
(DOCX)

**S10 Table. NCBI data submissions related to this study.**
(DOCX)

**S1 Fig. Genetic deletion of the *MSH2* gene and confirmation in JEC20a. (A)** Genetic deletion mutants lacking *MSH2* were engineered in JEC20**a** by replacing the *MSH2* open reading frame (ORF) with the dominant drug resistance marker *NAT* which confers resistance to nourseothricin via biolistic transformation. 5' and 3' UTRs are depicted as yellow boxes. Arrows depict locations of primers used to generate and verify *MSH2* deletion. The sets of primers denoted by gray arrows located upstream and downstream of the *MSH2* ORF were used to amplify flanking sequences homologous to the JEC20**a** *MSH2* endogenous locus to mediate homologous recombination (JOHE45551, 45552 and JOHE4555, 45556, respectively). Yellow arrows depict primers (JOHE45553 and JOHE45554) that amplified the *NAT* cassette and share homology with JOHE45552 and JOHE45556, respectively. The red arrows (JOHE45559,45560) and blue arrows (JOHE45822,45823) indicate primers that confirmed integration of the deletion allele and loss of the *MSH2* ORF, respectively. Gray lines indicate syntenic regions shared between the deletion allele and endogenous locus. **(B)** Gel electrophoresis of PCR products was used to confirm integration of a single copy of the *NAT* gene at the correct locus and that the wild-type *MSH2* gene was absent in the JEC20**a** mutant strains. (Spanning: JOHE45559,45560; WT in-gene: JOHE45822,45823; WT 5' junction: JOHE45559,45823; Δ5' junction: JOHE45559;45554; WT 3' junction: JOHE45822,45560; Δ3' junction: JOHE45553,45560).
(TIF)

**S2 Fig. Hypermutator phenotypes of JEC20a *msh2Δ* mutants on 5-FOA and papillation assays on 5-FOA, 5FC, and 5FU. (A)** Fluctuation analysis on YNB+5-FOA medium was performed in a similar manner to the analysis described in Fig 1A. **(B-D)** Representative images of plates used in papillation assays with independent JEC20**a** *msh2Δ* mutants and a KN99α *fur1Δ* mutant on **(B)** YNB+5-FOA medium, **(C)** YNB+5FC medium, and **(D)** YNB+5FU medium with the JEC20**a** parental strain and a KN99α *msh2Δ* mutant as controls. Strains were incubated on YNB+5-FOA medium for 6 days at 30°C before imaging. Strains were incubated

on YNB+5FC and YNB+5FU media for 3 days at 30˚C before imaging.
(TIF)

**S3 Fig. Germination frequencies of intraspecific progeny.** Average germination frequencies of progeny derived from **(A)** *C. neoformans* H99α x *C. neoformans* KN99**a** wild-type, unilateral *msh2Δ* (KN99α *msh2Δ* x KN99**a**), and bilateral *msh2Δ* crosses (KN99α *msh2Δ* x KN99**a** *msh2Δ-1* and KN99α *msh2Δ* x KN99**a** *msh2Δ-2*), and **(B)** *C. deneoformans* JEC21α x *C. deneoformans* JEC20**a** wild-type, unilateral *msh2Δ* (JEC21α x JEC20**a** *msh2Δ-1*), and bilateral *msh2Δ* crosses (JEC21α *msh2Δ-1* x JEC20**a** *msh2Δ-1* and JEC21α *msh2Δ-2* x JEC20**a** *msh2Δ-1*). Error bars represent standard error of the mean. Statistical significance was determined with one-way ANOVA and Tukey's post hoc test. * indicates $p<0.05$, ** indicates $p<0.01$, and *** indicates $p<0.001$.
(TIF)

**S4 Fig. PCR analysis of mating-type allele inheritance in *C. neoformans* x *C. deneoformans* hybrid progeny.** Sequence-specific primers for the mating-type (*MAT*) locus gene *STE20*, which can differentiate between *C. neoformans MAT***a**, *C. neoformans MAT*α, *C. deneoformans MAT***a**, and *C. deneoformans MAT*α, were used to characterize which *MAT* alleles each of the hybrid progeny inherited.
(TIF)

**S5 Fig. FACS analysis of *C. neoformans* x *C. deneoformans* hybrid progeny.** The *C. neoformans* strain H99α was used as a 1*n* haploid control and the *C. deneoformans* strain XL143 [65] was used as a diploid 2*n* control.
(PDF)

**S6 Fig. FACS analysis of *C. neoformans* x *C. neoformans* and *C. deneoformans* x *C. deneoformans* intraspecific progeny.** The *C. neoformans* strain H99α was used as a 1*n* haploid control and the *C. deneoformans* strain XL143 [65] was used as a diploid 2*n* control.
(TIF)

**S7 Fig. Self-filamentation in *C. neoformans* x *C. deneoformans* hybrid progeny.** Filamentation of hybrid progeny selected for whole-genome sequencing on MS medium after incubation for 14 days. AI187 is a self-filamentous, stable diploid *C. neoformans* strain [103] and served as a positive control for production of hyphae. Scale bars represent 100 μm.
(PNG)

**S8 Fig. Whole-genome comparisons of *C. neoformans* H99α and *C. deneoformans* JEC21α strains. (A)** Workflow to assess the genomic contribution of each parental species in the hybrid progeny. The sppIDer pipeline uses short-read sequencing data and a combined genome built from reference genomes of the two parental *Cryptococcus* species. **(B)** Dot-plot comparing the H99α assembly with the JEC21α reordered and reoriented assembly. Blue and red lines represent sequences with high similarities in the same and reverse orientations, respectively. **(C)** Linear plots showing overall synteny between the H99α and JEC21α genomes. The chromosomal positions of centromeres and the *MAT* locus are indicated by black and yellow bars, respectively. Chromosomes of JEC21α are color coded based on their synteny with the H99α chromosomes. Three major gross chromosomal changes previously documented distinguishing the two strains correspond to color changes within the same chromosome: TR indicates a reciprocal chromosomal translocation; INT indicates an introgression of a 14-gene region from *C. neoformans* to *C. deneoformans* that was mediated by transposable elements common to both lineages [27]; and SD indicates a segmental duplication following a nonreciprocal translocation involving the subtelomeric regions of JEC21 chromosomes 8 and

12 that presumably occurred during the construction of the congenic strain pair JEC21α/ JEC20**a** [104]. Chromosomal inversions are not indicated except for a large inversion on Chr3 of JEC21α (dashed arrow); see [42] for more detailed descriptions of inversions.
(TIF)

**S9 Fig. Nuclear genome composition of additional *C. neoformans* x *C. deneoformans* hybrid progeny. (A)** H99α x JEC20**a**. **(B)** H99α x JEC20**a** *msh2Δ-1* **(C)** KN99α *msh2Δ* x JEC20**a** **(D)** H99α *msh2Δ* x JEC20**a** *msh2Δ-1*. For each progeny, read-depth plots (normalized to the genome-wide average coverage) are colored according to each parental species contribution as shown in the key on the top right, and a schematic representation of the inferred karyotype is depicted on the right. Homeologous chromosomes are color coded based on the H99 reference (see S8 Fig for details) and asterisks in JEC21α indicate chromosomes in reverse-complement orientation. Red arrowheads mark recombination breakpoints between homeologous chromosomes and/or loss of heterozygosity (also highlighted by red boxes in the karyotype panels). Where detected, the breakpoints of additional recombination events within the same chromosome are indicated by light blue arrowheads. Circular black labels: (a) marks changes in ploidy in a subset of the population of cells that were sequenced; (b) marks chromosome breakage events repaired by *de novo* telomere addition (see S13 Fig for details); (c) indicates recombination events next to the *MAT* locus; (d) marks a break at the rDNA locus; (e) marks a complex chromosomal aberration that cannot be explained by simple rearrangements and required further investigation.
(PDF)

**S10 Fig. Aneuploidy of the *C. neoformans* x *C. deneoformans* hybrid progeny.** Graphs depicting the number of hybrid progeny with aneuploidies for each chromosome **(A)** and genetic cross **(B)**. Graph showing no correlation between whole-chromosome aneuploidy events observed (y-axis) and chromosomal size (x-axis). Blue line represents linear fit and blue shaded area represents the 95% confidence interval for the fitted line.
(TIF)

**S11 Fig. Quantification of heterozygosity across genomes of *C. neoformans* x *C. deneoformans* hybrid progeny.** Plot showing the percentage of heterozygosity for each individual progeny (represented by different dots) grouped by type of cross (see S5 Table for details). The horizontal red line depicts the mean heterozygosity values. Genomes of progeny derived from bilateral *msh2Δ* × *msh2Δ* mutant crosses were significantly more heterozygous only when compared to hybrid progeny derived from wild-type H99α x JEC20**a** crosses (Statistical analysis: Kruskal-Wallis test, followed by Dunn's test, $p<0.05$).
(PDF)

**S12 Fig. The progeny of intraspecific crosses of *C. neoformans* and *C. deneoformans* or their derived *msh2Δ* mutants are predominantly haploid.** Read depth (binned in 1-kb non-overlapping windows) was plotted along each chromosome of *C. neoformans* H99 **(A)** and *C. deneoformans* JEC21 **(B)** to screen for chromosome aneuploidy. For each sequenced strain, read depth was normalized to the median read depth for that strain, log2-transformed, and plotted as a heat map in IGV viewer. Ploidy was also measured by FACS and the results indicate that progeny #5 of KN99α *msh2Δ* × KN99**a** is mostly diploid except for chromosome 13 ($2n$ -1), and progeny #3 and #4 of KN99α *msh2Δ* × KN99**a** *msh2Δ-2* have gained additional copies of chromosomes 2 and 7, respectively ($1n$ + 1). The asterisk indicates that the biased sequence coverage observed along Chr2 of progeny 3 from the KN99α *msh2Δ* × KN99**a** *msh2Δ-2* cross might be due to biochemical effects related to library preparation or

sequencing.
(TIF)

**S13 Fig. Chromosome breaks in the hybrid progeny can be repaired by *de novo* telomere addition.** Breakpoints were detected in different chromosomal locations (see S7 Table for details), including *CEN14* and *CEN9* of JEC21α **(A and B)**, a T1 transposable element located at the end of Chr6 of JEC21α **(C)**, or in other genomic locations **(D–F)**. Each panel shows the result of read mapping for one or more progeny that underwent chromosome breakage and healing via *de novo* telomere addition (strains names in boldface type) and a control strain in which no breaks were detected on the same region (strain names in normal font type). Breakage and *de novo* telomere addition was inferred, respectively, by abrupt changes in read coverage (depicted as bars on the top and colored as shown in the key) and by the presence of reads with telomeric repeats at the breakpoints. When two copies of the same chromosome are present, only a subset of reads are expected to contain telomeric repeats (as shown e.g. in panel F).
(PDF)

**S14 Fig. Events of chromosome break and repair via *de novo* telomere addition in intraspecific crosses of *C. neoformans* and *C. deneoformans* are rare.** Breakage and *de novo* telomere addition was inferred, respectively, by abrupt changes in read coverage (depicted as bars on the top and colored as shown in the key) and by the presence of reads with telomeric repeats at the breakpoints. Such events were not detected in any of the 19 *C. deneoformans* sequenced intraspecific progeny and were found in only 1 progeny (#3 of KN99α *msh2Δ* × KN99**a** *msh2Δ-2*) out of 19 progeny derived from the *C. neoformans* intraspecific crosses. In this strain, a region of ~17 kb, which contained a few putative genes and predicted transposable elements, was deleted from the 5' end of chromosome 13.
(TIF)

## Acknowledgments

We thank Tom Petes, Sue Jinks-Robertson, and Paul Magwene for critical reading and comments on the manuscript. We thank Kevin Zhu for critical reading, comments on the manuscript, and generating figures. We thank Jay Jawahar for his assistance in the initial phases of constructing the JEC20**a** *msh2Δ* mutants. We also thank the Madhani Laboratory for the KN99α *msh2Δ* and *fur1Δ* deletion strains.

## Author Contributions

**Conceptualization:** Shelby J. Priest, Joseph Heitman.

**Data curation:** Marco A. Coelho.

**Formal analysis:** Marco A. Coelho, Verónica Mixão.

**Funding acquisition:** Shelby J. Priest, Joseph Heitman.

**Investigation:** Shelby J. Priest, Marco A. Coelho, Verónica Mixão, Shelly Applen Clancey, Yitong Xu, Sheng Sun.

**Methodology:** Shelby J. Priest, Marco A. Coelho, Verónica Mixão, Shelly Applen Clancey, Toni Gabaldón.

**Project administration:** Shelby J. Priest, Joseph Heitman.

**Resources:** Shelby J. Priest, Yitong Xu, Sheng Sun, Joseph Heitman.

**Software:** Verónica Mixão.

**Supervision:** Joseph Heitman.

**Validation:** Shelby J. Priest, Marco A. Coelho, Verónica Mixão, Sheng Sun.

**Visualization:** Shelby J. Priest, Marco A. Coelho, Shelly Applen Clancey.

**Writing – original draft:** Shelby J. Priest, Marco A. Coelho, Verónica Mixão, Shelly Applen Clancey.

**Writing – review & editing:** Shelby J. Priest, Marco A. Coelho, Verónica Mixão, Shelly Applen Clancey, Yitong Xu, Sheng Sun, Toni Gabaldón, Joseph Heitman.

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
