## [Decision Letter · Decision Letter 0]

8 Jun 2020

Dear Dr Priest,

Thank you very much for submitting your Research Article entitled 'Factors enforcing the species boundary between the human pathogens Cryptococcus neoformans and Cryptococcus deneoformans' to PLOS Genetics. Your manuscript was fully evaluated at the editorial level and by three independent peer reviewers with expertise in the field. The reviewers appreciated the attention to an important problem, but raised some substantial concerns about the current manuscript. Based on the reviews, we will not be able to accept this version of the manuscript, but we would be willing to review a revised version. We cannot, of course, promise publication at that time.

If you decide to revise the manuscript for further consideration at PLOS Genetics, please aim to resubmit within the next 60 days, unless it will take extra time to address the concerns of the reviewers, in which case we would appreciate an expected resubmission date by email to plosgenetics@plos.org.

[LINK]

We are sorry that we cannot be more positive about your manuscript at this stage. Please do not hesitate to contact us if you have any concerns or questions.

Yours sincerely,

Gregory P. Copenhaver

Editor-in-Chief

PLOS Genetics

Gregory Barsh

Editor-in-Chief

PLOS Genetics

Reviewer's Responses to Questions

**Comments to the Authors:**

Reviewer #1: In this study the authors performed crosses between two species of Cryptococcus (C. neoformans and C. deneoformans, ~15% divergent), and then analyzed progeny that were derived from wild-type or mismatch repair defective (msh2 null, unlateral or bilateral) parents. They found that progeny derived from msh2null bilateral crosses showed higher viability and levels of aneuploidy. Recombination events were seen at low frequency in the progeny (1 to 1.5 events per cross), with similar frequencies in wild-type and msh2null crosses. Based on these observations, the authors suggest that the mismatch repair system is not playing a role in limiting meiotic recombination between divergent Cryptococcus species.

In general, the work is well done and clearly explained and shows a role for the mismatch repair factor Msh2 in enforcing species boundaries in Cryptococcus. I have two major comments indicated below.

1. The authors do a nice job describing the previous work in bacteria (Hfr crosses between E. coli and Salmonella typhimurium) and yeast (meiotic recombination involving interspecific hybrid of S. cerevisiae and S. paradoxus) indicating that the mismatch repair system acts to maintain species barriers. However, in addition to these studies, it’s also worth thinking about the following:

Elliott and Jasin (MCB 21:2671) measured I-SceI induced recombination between divergent substrates in mammalian cells. They observed that substrates with 1.5% divergence displayed 10-fold higher levels of recombination in Msh2-/- compared to wild-type cells. In addition, the Msh2-/- recombinants displayed a much higher level of uncorrected tracts, indicating that many recombination events involved the formation of heteroduplex DNA.

Anand et al. (Nature 544:377) examined break-induced replication in yeast using templates that contained different densities of DNA polymorphisms. A key observation of their work was that “When recombination occurs without a protruding nonhomologous 3' tail, the mismatch repair protein Msh2 does not discourage homeologous recombination. However, when the DSB end contains a 3' protruding nonhomologous tail, Msh2 promotes the rejection of mismatched substrates.” In addition, they showed that “Nearly all mismatch correction depends on the proofreading activity of DNA polymerase-δ, although the repair proteins Msh2, Mlh1 and Exo1 influence the extent of correction.”

Peterson et al. (Mol Cell doi: 10.1016/j.molcel.2020.04.009) examined meiotic recombination in F1 hybrid mice. They analyzed events at two recombination hotspots loci that in the hybrid displayed a high (~1%) polymorphism density. Interestingly, they observed no difference in the frequency of crossover events in wild-type and Msh2-/-, but still observed high levels of heteroduplex DNA in Msh2-/-.

I summarized the above studies because they involved systems where a DSB was induced/occurred at a specific site/region and polymorphisms were present that allowed them to monitor the presence of heteroduplex DNA. In addition, the studies illustrate the different outcomes that can be obtained when mismatch repair is ablated. In some cases, mismatch repair had no effect on the overall frequency, but did affect the repair of heteroduplex intermediates. With this said I have the following concerns with the work presented by Priest et al.:

A. Crosses between the two species yielded very low numbers of events (1 to 1.5 per cross), and as the authors stated on line 479, it is “difficult to determine if these recombination events occurred during meiosis or subsequent mitotic growth during the culturing required for whole genome sequencing.”

B. Their analysis did not map the exact breakpoints or determine whether there was a significant polymorphism density in the regions undergoing recombination, or if there was any evidence of uncorrected repair tracts. It was intriguing that the authors observed a fair number of de novo telomere addition events in msh2 bilateral crosses, perhaps indicative of an increase in breakage under these conditions. Because it’s not clear where the breakpoints occurred, it’s hard to determine if the breaks occurred in repetitive regions where one might not expect to see effects of mismatch repair. These issues make it difficult for me conclude that the authors can provide mechanistic evidence for factors enforcing the species boundary between different Cryptococcus species.

2. The authors report low rates of loss of heterozygosity and high rates of aneuploidy in hybrid progeny from msh2null mutant crosses. This is an interesting observation but, at least for this reviewer, not a lot of mechanistic insights were provided. It would be interesting to perform crosses within the same species (C. neoformans x C. neoformans, C. deneoformans x C. deneoformans), analyzing progeny derived from both wild-type and msh2null isolates. This would provide information on whether some of the phenotypes that were observed were the result of sequence divergence or other activities promoted by Msh2.

Other comments

1. Line 70: Define what you mean by “transgressive. A violation of accepted dogma?

2. Lines 109-121. The authors provide many details about how many hybrids exist in nature, but they don’t address how often they are fertile. This paragraph is nicely written, but feels like a distraction from the topic they are addressing (factors that enforce species boundaries).

3. Line 197. It would be worth explaining here what the targets are in the rapamycin +FK506 (FRR1 is a common target) and 5-FOA (URA3 or URA5) assays that lead to resistance to these drugs. This would give the reader an immediate context in terms of the kinds of mutations that can be identified.

4. The detailed analysis of the msh2null-1 mutant feels like a distraction. It’s somewhat interesting that the authors could track down why msh2null-1 failed to produce 5-FOA resistant isolates, but the extended analysis of the mutant does not provide much to the overall story. I see no reason why it can’t be presented in the Materials and Methods or supplement, rather than being emphasized in the Results and Discussion.

5. Line 253 and elsewhere. It would be valuable in the text to provide some of the statistical analysis presented in the Figures that indicated that the differences seen between wild-type and msh2null were significant.

6. Line 491. Typo-do you mean 15% divergent sequences?

Reviewer #2: Priest et al. aim to identify the role of the gene MSH2 in hybrid sterility in the pathogenic fungi Cryptococcus. Previous work in Saccharomyces has shown that anti-recombination due to sequence divergence is responsible for sterility between Saccharomyces species. Knocking out MSH2 can rescue part of this sterility by increasing the number of meiotic events between divergent sequence and thus promoting proper chromosome segregation. Here, the researchers show that MSH2 does function in DNA MMR in Cryptococcus similar to other organisms, but does not increase successful recombination in hybrids. This is at odds with their results that germination frequency is greatly increased, and it is unclear what mechanism is leading to higher viability. They also demonstrate other interesting results, like high rates of diploidy in the viable hybrids, and potentially an ability for hybrids to self fertilize. They thus show support that MSH2 has diverged in function in Cryptococcus, surprising as its role in anti-recombination is conserved from bacteria to eukaryotes.

I think the topic of hybridization in fungal pathogens is really interesting, and a better understanding of the mechanisms and phenotypes associated with hybrid viability is needed. I particularly enjoyed the Discussion section where this was addressed. I have several larger questions and comments, below, and some clarifying questions I think would lead to a more clear manuscript.

Throughout the manuscript there is a lot of reference to phenotypes “unique to hybrid progeny,” but it’s unclear if these are unique to hybrid progeny because non-hybrid crosses are not analyzed (or at least not discussed). For example, is the ability to undergo sexual reproduction after 3 weeks under abnormal conditions unique to msh2 hybrids? What do wild-type hybrid and non-hybrid cross controls look like? What are the difference between hybrids and non-hybrids under the same conditions after this time period? Statistics are needed to identify if differences are significant. Similarly, the observation of repairs of double strand breaks with telomeric repeat sequences (Lines 341-345) - is this unique to hybrid progeny with msh2 or does this happen in msh2 knockouts in non-hybrid progeny as well?

I’m not sure I agree with the statement that loss of Msh2 did not relax species boundaries in Cryptococcus. It seems from their results of germination frequency, that MSH2 does indeed play a role in maintaining species boundaries (when msh2 is deleted in both species, germination increases), in line with results from other organisms. The statement that alternative pathways limit recombination is also unclear, as only MSH2 was tested in this capacity and the MMR pathway still appears to be functioning in many aspects of chromosome segregation/meiosis.

Could you quantify the aneuploidy you see? It would be helpful to see proportions of crosses with aneuploidy across non-hybrids, non-hybrid msh2, msh2 knockouts in wild type hybrids, unilateral and bilateral msh2 hybrids. Is the aneuploidy random or are there particular chromosomes that are lost or gained from one species?

Is recombination required for successful completion of meiosis in Cryptococcus? What is the expected meiotic recombination frequency? It’s particularly interesting that the few recombination events identified are in the wild-type hybrid crosses instead of the msh2 knockouts. What might this suggest?

Other comments and questions

What was the intention behind creating a series of msh2 knockouts and why do the independent msh2 knockouts have variable mutation rate and other phenotypes? Is this likely because of mutations at other loci due to hypermutator phenotype?

Can you describe what you mean by self-fertility? To a naïve Cryptococcus person, I don’t know what the typical reproductive cycle is. Can a (non-hybrid) diploid not self-fertilize normally? Can they persist asexually?

On a related note, I wasn’t clear what the expected ploidy of the hybrid progeny was until the discussion.

Are the hybrids isolated from environmental/clinical samples typically diploid?

What is wild type germination frequency between strains? (non hybrid cross) What about in msh2 knockouts in non-hybrids?

Figure 2A and text: Do bald basidia = sterility? Hard to tell because the PDF version of the figures is pretty blurry, but I can’t distinguish differences between panels. Could you point out the relevant mating structures (or lack there of) that I should be looking for? Is the relevant point that even though there are more bald basidia that there is higher germination?

Did you look at the germination frequency of the F1 hybrid progeny that produced sexual structures after incubation on YPD for 3 weeks?

Some other literature to include:

Bozdag et al. bioRXiv https://doi.org/10.1101/755165

Rogers et al. 2018 https://doi.org/10.1371/journal.pbio.2005066

Li et al. 2019 https://doi.org/10.1038/s41467-019-12927-7

Reviewer #3: The manuscript by Priest et al. explores the role of MSH2 and mismatch repair (MMR) in the post-zygotic reproduction barrier in hybrids between two Cryptococcus species. Hybrids between C. neoformans and C. deneoformans are readily found in the environment and in higher frequency in clinical cases. Like the Saccharomyces yeasts, there appears to be little or no pre-zygotic reproductive isolation, however these hybrids are generally sterile with relatively low germination frequencies of meiotic products, a post-zygotic barrier. The manuscript tests the role of MMR in this low fertility by knocking out MSH2 which has been shown to improve meiotic spore viability in Saccharomyces hybrids by increasing crossing over and improving segregation of chromosomes. They find that knocking out MSH2 in both species parent of hybrids increases fertility significantly, however this does not appear to be via increased crossing over during meiosis. The authors conclude that other factors must be involved in repressing meiotic recombination in these Cryptococcus hybrids.

In general, the experiments are well done and the conclusions reasonably sound. I think the authors are underselling the role of MSH2 which clearly is involved in some way in the reproductive isolation barrier and some further discussion of this is merited. The presumed role of MMR in reproductive isolation in hybrids, based on experiments in bacteria (E. coli and S. typhimurium) and Saccharomyces (mostly S. cerevisiae and S. paradoxus with a similar level of diversity as seen here in the Cryptococcus species), is to be saturated for mismatches during strand invasion at the beginning of the recombination process, resulting in abortion of the initial recombination intermediates. This has the consequence of reducing meiotic recombination and increasing meiosis I non-disjunction due to the lack of tension on the spindle. It’s clear from the Saccharomyces studies that different MMR components may have independent roles in this barrier – pms1 and msh2 mutants in CHR III homeologous XO (Chambers et al. 1996) – despite their shared role in MMR in vegetative cells. MSH2 has also been shown to have a mismatch independent role in telomere repeat recombination which may or may not be relevant here but indicates additional roles independent of sequence divergence. As the role in Saccharomyces and bacteria is binding mismatches in strand invasion structures and perhaps aborting them, and not in XO resolution, then it might be that in Cryptococcus the lack of MSH2 is allowing meiosis to proceed far enough to generate viable products without the resolution of interactions into XOs. Perhaps these invasions are resolved as non-crossovers, even after meiosis is completed. Given that the progeny of the bilateral msh2 hybrids are near diploid hybrids themselves, meiosis appears to result in a skipping of the reductional division which might occur when all the homeologous chromosomes are linked by strand invasions. They might not be resolvable as XOs or if they are, they may be inviable (an explanation of the reduced frequency of meiotic structures in these mutant hybrids?) such that only those that do not complete recombination are viable. This is just speculation of course and not easily addressed experimentally here for this paper, but the authors should address the issue that MSH2 does have a role in the post-zygotic barrier, but it is an unknown or unexpected role. I think this is more exciting than the conclusion that there are other repressors of meiotic recombination.

More specific comments:

The near haploid and near diploid nature of all the progeny assessed makes sense from the assessment of the tolerance of aneuploids in S. cerevisiae (Parry and Cox 1970 Genet Res 16) where viable segregants from a triploid cross were all close to euploidy. The data presented here looks similar with the wt cross exhibiting both, the heterozygous msh2 crosses exhibiting various levels of heterozygosity in the near diploid progeny with lower levels consistent with aneuploidy (2n-1). The progeny of the msh2 homozygous cross appear to be near diploid with whole complements of each parental species which is different than random segregation of chromosomes and look more like Meiosis I non-disjunction. One question – in the 3 near haploid progeny of the wt cross are the chromosomes present a mixture of each parental species?

The lack of detections of any possible increased recombination could be due to various factors. Given that the homozygous msh2 progeny were all near diploid and heterozygous across the genome, the authors did the right thing to look for cryptic XOs (in just the sequence) by looking for reads that crossed breakpoints. In general, when XOs are resolved they will be in the longer tracts of homology. Depending on the distribution of the SNPs in the two species and the range of homologous tract lengths (are there any of 200 bp long for example?), it is possible that most XOs will be in tracts longer than the reads could cover and therefore such recombination events would be missed. Alternatively, it is possible that the recombination events lead to death as is the case in S. cerevisiae where one partner of two recombining chromosomes is unresolved (Chambers 1996) resulting in a dead spore while the other partner survives. This isn’t consistent with the results presented here but all viable progeny here appear to be MI non-disjunctions which may hide such events by resolving them after meiosis.

Line 430 – If there is little or no recombination then QTL mapping will be difficult.

Line 491 – it’s 15% divergent not 85% divergent

Line 514 – loss of MSH2 clearly did reduce the boundary between these species as there was an increase in viability of meiotic products – many of these displayed transgressive sexual behaviour as well, meaning that it may be heritable. Question – were basidiospores dissected from these progeny? How viable were they?

**Have all data underlying the figures and results presented in the manuscript been provided?**

Reviewer #1: Yes

Reviewer #2: Yes

Reviewer #3: Yes

PLOS authors have the option to publish the peer review history of their article (what does this mean?). If published, this will include your full peer review and any attached files.

Reviewer #1: No

Reviewer #2: No

Reviewer #3: No

---

## [Decision Letter · Decision Letter 1]

4 Dec 2020

Dear Dr Priest,

We are pleased to inform you that your manuscript entitled "Factors enforcing the species boundary between the human pathogens Cryptococcus neoformans and Cryptococcus deneoformans" has been editorially accepted for publication in PLOS Genetics. Congratulations!

The reviewers were particularly complimentary of the careful revisions you made - bravo!  As you will see (below) two of the reviewers had some final small suggestions that you should consider as you prepare your final draft for the production team (the editorial team will not need to re-evaluate).

Yours sincerely,

Gregory P. Copenhaver

Editor-in-Chief

PLOS Genetics

Gregory Barsh

Editor-in-Chief

PLOS Genetics

Comments from the reviewers (if applicable):

Reviewer's Responses to Questions

**Comments to the Authors:**

Reviewer #1: I am impressed by the efforts made by the authors to address my concerns, especially their analysis of intraspecific crosses. I feel that this new work provides a better understanding of the phenotypes seen in the msh2 hybrid crosses. It also helped me better understand the Discussion (beginning on line 489) where the authors discussed why the msh2null mutant hybrid progeny displayed higher ploidy levels compared to wild-type. The possibility of divergent chromosomes being linked by strand invasions is an interesting idea, but then it requires some interesting explanations-they are resolved either through non crossovers or crossovers, with crossovers resulting in inviability. This will clearly require some sorting out in the future. With that said, I am curious what might be found in future experiments in sgs1 and mph1 mutant backgrounds.

Lastly, I find it curious that the hybrid progeny in msh2 remain aneuploid and as shown in S4 Table it appears that they show 2N or 4N content rather than 1N. If I understand this correctly it would make it difficult to identify heteroduplex DNA in the progeny of the random spores dissection. Perhaps adding a short sentence to this effect would be valuable (unless of course I interpreted this incorrectly!).

Reviewer #2: I thought Priest et al did a fantastic job responding to my previous comments and questions. The added experiments, analyses, figures, and tables greatly improve the manuscript. I think their strengthened results regarding the production of hyphae by hybrids under abnormal conditions (Figure 3B and TableS3) are really exciting and interesting. I really appreciated the thoughtful responses and clear effort that went into addressing reviewer feedback.

Minor comments:

Lines 210-214: Include statistics

Lines 264-278: Any idea why the bilateral intra-species germination frequencies are so different between the species? (a bilateral msh2Δ C. neoformans intraspecific cross germinated on average only 71% of the time; progeny from bilateral msh2Δ C. deneoformans intraspecific crosses germinated only 36% of the time)

Reviewer #3: I am satisfied with the responses to my and other reviews.

**Have all data underlying the figures and results presented in the manuscript been provided?**

Reviewer #1: Yes

Reviewer #2: Yes

Reviewer #3: Yes

PLOS authors have the option to publish the peer review history of their article (what does this mean?). If published, this will include your full peer review and any attached files.

Reviewer #1: No

Reviewer #2: No

Reviewer #3: No

**Data Deposition**

http://datadryad.org/submit?journalID=pgenetics&manu=PGENETICS-D-20-00781R1

**Press Queries**

---

## [Editor Report · Acceptance letter]

14 Jan 2021

PGENETICS-D-20-00781R1 

Factors enforcing the species boundary between the human pathogens Cryptococcus neoformans and Cryptococcus deneoformans 

Dear Dr Priest, 

We are pleased to inform you that your manuscript entitled "Factors enforcing the species boundary between the human pathogens Cryptococcus neoformans and Cryptococcus deneoformans" has been formally accepted for publication in PLOS Genetics! Your manuscript is now with our production department and you will be notified of the publication date in due course.

With kind regards,

Melanie Wincott

PLOS Genetics

On behalf of:
